# Injecting Inductive Bias to 3D Gaussian Splatting for Geometrically Accurate Radiance Fields

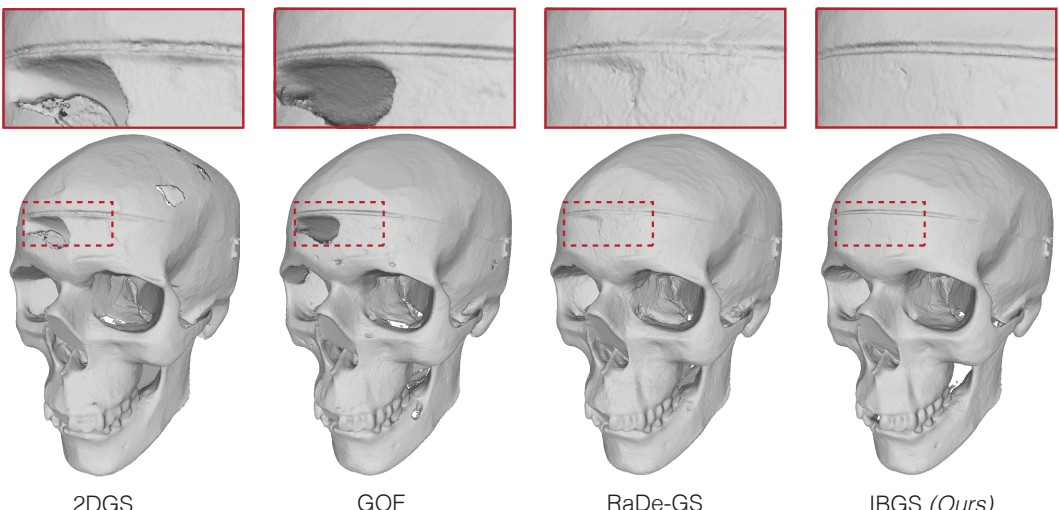

2DGS        GOF        RaDe-GS        IBGS *(Ours)*

Figure 1: Our approach enhances geometric accuracy of 3D Gaussian Splatting by parameterizing covariance, the key determinant of surface normal, with distribution of adjacent Gaussians. Zoom in for detailed comparison.

## ABSTRACT

3D Gaussian Splatting (3DGS) has significantly advanced high-fidelity, real-time novel view synthesis. However, its discrete nature limits the accurate reconstruction of geometry. To address this issue, recent methods have introduced rendering and regularization of depth and normal maps from 3D Gaussians, leading to plausible results. In this paper, we argue that computing normals from independently trainable Gaussian covariances contradicts the strict definition of normals, which should instead be derived from the distribution of neighboring densities. To address this, we introduce an inductive bias into 3DGS by explicitly parameterizing covariances of Gaussians using principal axes and variances of distribution computed from neighboring Gaussians. These axes and variances are then regularized to ensure local surface smoothness. Our approach achieves state-of-the-art performance on multiple datasets.

## 1 INTRODUCTION

Along with the advancements made by Neural Radiance Fields (NeRF) (Mildenhall et al., 2021), 3D Gaussian Splatting (3DGS) (Kerbl et al., 2023) has significantly influenced novel view synthesis research, enabling high-fidelity and real-time rendering from a collection of posed 2D images. Progress in NeRF research naturally led to advances such as NeuS (Wang et al.) and Neuralangelo (Li et al., 2023) for accurate geometry representation via implicit Signed Distance Fields (SDF). Similarly, the evolution of geometrically accurate Gaussian Splatting is ongoing by recent works such as 2D Gaussian Splatting (Huang et al., 2024), Gaussian Opacity Fields (GOF) (Yu et al., 2024b), and RaDe-GS (Zhang et al., 2024).

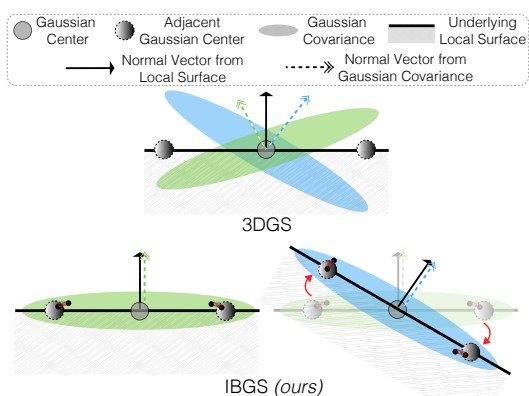

GT  RGB  Depth  Normal

Figure 2: **Renderings on scenes trained with IBGS on MipNeRF-360 (*top*) and DTU (*bottom*) dataset.** IBGS is a model for geometrically accurate representation of 3D Gaussians.

The core insight of the recent 3DGS-based geometry reconstruction methods is to determine the depth and normal of a Gaussian given its covariance. For instance, 2DGS defines a surfel disk, so that a normal can be defined as the vector perpendicular to the disk. Meanwhile, GOF and RaDe-GS calculate the intersection plane between 3D Gaussian and a camera ray, so that normal of the plane can be regarded as the normal of the Gaussian.

However, all these methods calculate normal from a covariance matrix independently trainable for each Gaussian. In fact, this schematics is counter-intuitive to the common understanding of normal estimation in computational graphics (Shirley et al., 2009), where normal is instead defined by the gradient of adjacent density distribution. Consider 3DGS in Figure 3. From the distribution of the two adjacent Gaussians located left and right, a geometrically reasonable surface normal is the vector perpendicular to the local surface formed by these Gaussians.

Figure 3: Normal determined by fully trainable covariance in 3DGS does not necessarily correspond to normal of local surface determined by neighboring Gaussians. However, covariance in IBGS is directly parameterized using the neighboring Gaussians, so that the two normal vectors are aligned. Also, updating covariance leads to the relocation of adjacent Gaussians in IBGS.

However, a fully trainable covariance in 3DGS can be oriented arbitrarily, creating a discrepancy between the normal calculated from the covariance and the normal calculated from the underlying surface formed by adjacent Gaussians. In other words, the orientation of covariance has room to fall into local minima while satisfying the photometric loss, which is ill-posed for accurate geometry reconstruction.

In this paper, we tackle this issue by directly injecting **I**nductive **B**ias to 3D **G**aussian **S**platting (**IBGS**). Specifically, we parameterize the covariance matrix of a Gaussian using the distribution of adjacent Gaussians as illustrated in the bottom left figure of Figure 3. Our design enforces a Gaussian covariance to be oriented by the locations of the adjacent Gaussians via Singular Value Decomposition (SVD). Thus, updating covariance requires back-propagating gradients to adjacent Gaussian locations, as illustrated in the bottom right figure of Figure 3. As these adjacent points also form covariances using their own neighbors, the scene eventually forms a set of globally chained Gaussians by satisfying our inductive bias. To form a coherent and smooth local surface of this graph-like structure, we also propose geometry regularization methods to *(i)* induce the neighboring Gaussians to be distributed along a local plane and *(ii)* assimilate normals of adjacent local planes to form a smooth surface. As a result, our method yields high-quality rendering of scene geometry such as depth and normal reported in Figure 2, enabling high-quality surface reconstruction results compared to state-of-the-arts as shown in Figure 1.

In summary, our contribution is three-fold:

○ We propose a novel variant of a 3DGS model, which parameterizes a covariance of a Gaussian with neighboring Gaussians, to reflect local geometry structure to the Gaussian covariance.

○ We propose regularization losses for smooth and coherent surface reconstruction.

○ Our method achieves competitive quantitative and qualitative results in surface reconstruction among 3DGS-based methods, while maintaining faster training time compared to implicit neural representation-based methods.

## 2 RELATED WORKS

**Novel View Synthesis** Given a set of posed images, NeRF (Mildenhall et al., 2021) employs a multi-layer perceptron (MLP) to model the density and view-dependent color of a scene. Image is rendered from the MLP via the volume rendering frameworks (Drebin et al., 1988; Kajiya & Von Herzen, 1984), and optimized with a photometric loss for training. Following the broad impact of NeRF on the research community, advancements have been made to enhance its efficiency and effectiveness. Notably, several approaches have accelerated NeRF's training and rendering process through the integration of octree (Yu et al., 2021), sparse voxel-grid (Fridovich-Keil et al., 2022), multi-resolution hash-grid (Fridovich-Keil et al., 2022), and factorized tensors (Chen et al., 2022). Other lines of work successfully mitigate aliasing effect (Barron et al., 2021; 2023; Hu et al., 2023), or model unbounded scenes (Zhang et al., 2020; Barron et al., 2022). More recently, 3DGS (Kerbl et al., 2023) has emerged as a powerful technique for achieving real-time and high-fidelity rendering. Representing scenes as a set of 3D Gaussians, the method employed EWA volume splatting (Zwicker et al., 2001) to rasterize the 3D Gaussians into screen space. Subsequent research has refined this approach by improving rendering quality through anti-aliasing (Yu et al., 2024a) or extending to dynamic scene modeling (Wu et al., 2024; Yang et al., 2024).

**Surface Reconstruction** Traditionally, Multi-View Stereo (MVS) techniques (Schönberger et al., 2016; Yao et al., 2018; Yu & Gao, 2020) addressed the 3D reconstruction problem from multi-view images. These methods involve a series of steps, including feature matching, depth map estimation, and the fusion of these maps into a point cloud. The point cloud is then used to reconstruct a surface, often via approaches like screened Poisson surface reconstruction (Kazhdan & Hoppe, 2013).

Meanwhile, NeRF-based methods have been employed for accurate surface reconstruction, as novel view synthesis from implicit scene representation is closely related to underlying 3D reconstruction. For instance, pioneering methods such as UNISURF (Oechsle et al., 2021), NeuS (Wang et al.), and VolSDF (Yariv et al., 2021) propose an implicit SDF for surface representation, followed by its volume rendering for supervision with posed images. Triangle meshes can then be straightforwardly extracted using techniques like Marching Cubes (Lorensen & Cline, 1998). Noticeable progress in this field has been driven by Neuralangelo (Li et al., 2023), which proposed a regularization method with numerical gradient and coarse-to-fine representation for SDFs. Other lines of work integrated monocular priors (Yu et al., 2022) or sensor depth (Azinović et al., 2022) to integrate geometric cues for accurate surface representations.

Although 3DGS has recently gained traction as a robust method for novel view synthesis, extracting 3D surfaces from Gaussian splats presents significant challenges. To address this issue, methods such as SuGaR (Guédon & Lepetit, 2024) and NeuGS (Chen et al., 2023) have introduced the concept of flat Gaussians, which are designed to better conform to object surfaces, thereby improving surface alignment and reconstruction quality. 2DGS (Huang et al., 2024) similarly proposes 2D Gaussian surfel, yet formulates the rasterization process into 2D-to-2D transformation, making the splatting process more perspectively accurate. Recent works such as GOF (Yu et al., 2024b) and RaDe-GS (Zhang et al., 2024) computes ray-Gaussian intersection to render depth and normal. Meanwhile, a line of works improves geometry by regularizing Gaussians' scale. Hyung et al. (2024) regularizes effective rank of Gaussian scales, while Hwang et al. (2024) minimizes scale along the surface normal estimated from monocular prior. However, none of them questions the importance of the relationship between the neighboring Gaussians for geometrically accurate representation.

# 3 METHOD

## 3.1 PRELIMINARIES

**3D Gaussian Splatting**   3DGS proposes scene representation and rasterization method from a collection of 3D Gaussian primitives. A 3D Gaussian primitive $\mathcal{G}$ contains a set of trainable parameters: the mean position $\boldsymbol{\mu} \in \mathbb{R}^{3\times1}$, a covariance matrix $\boldsymbol{\Sigma} \in \mathbb{R}^{3\times3}$, opacity $\alpha \in [0,1]$, and a color $c$ expressed in spherical harmonics to model view dependency. The covariance matrix $\boldsymbol{\Sigma}$ is positive semi-definite and is constructed using a rotation matrix $\mathbf{R} \in \mathbb{R}^{3\times3}$, which is parameterized by a quaternion, and a diagonal scale matrix $\mathbf{S}$. This Gaussian primitive in 3D space can be described as

$$\mathcal{G}(\mathbf{x}) = e^{-\frac{1}{2}(\mathbf{x}-\mu)^T \boldsymbol{\Sigma}^{-1} (\mathbf{x}-\mu)}. \tag{1}$$

The primitives are then projected onto a 2D plane (Zwicker et al., 2001). In this projection, the covariance matrix is transformed into screen space as $\boldsymbol{\Sigma}' = \mathbf{J}\mathbf{W}\boldsymbol{\Sigma}\mathbf{W}^\top\mathbf{J}^\top$, where $\mathbf{W}$ is the world-to-camera transformation matrix and $\mathbf{J}$ is the Jacobian of the affine projection matrix approximation. The final image is then produced by alpha-blending these Gaussians based on their depth as $\mathbf{c}(\mathbf{u}) = \sum_{m=1}^{M} c_m \alpha_m \mathcal{G}_m^{2D}(\mathbf{u}) \prod_{j=1}^{m-1}(1 - \alpha_j \mathcal{G}_j^{2D}(\mathbf{u}))$, where $\mathcal{G}^{2D}$ is the Gaussian projected to screen-space and $\mathbf{u}$ refers to the screen coordinate.

Given that the Gaussians are initially derived from sparse Structure-from-Motion (SfM) points, the Adaptive Density Control (ADC) technique is applied to refine and densify the scene during optimization. ADC clone and splits Gaussians (Yu et al., 2024b) when the densification signal of a Gaussian, $\frac{\partial L'}{\partial \mathbf{u}}$, is greater than a predefined threshold $\tau$ as following:

$$\frac{\partial L'}{\partial \mathbf{x}} := \sum_{i \in \mathcal{P}} \left\| \frac{\partial L}{\partial \mathbf{p}_i} \frac{\partial \mathbf{p}_i}{\partial \mathbf{x}} \right\|_2 > \tau, \tag{2}$$

where $\mathbf{x}$, $\mathcal{P}$, and $\mathbf{p}_i$ denote the center of projected Gaussians, the list of pixel indices that the Gaussian contributed to, and the following pixel, respectively. This condition ensures that regions with significant positional gradients, which indicate incomplete reconstruction, are effectively densified by adding more Gaussians, thereby improving the expressibility.

**Rasterization of Depth and Normal**   Given a Gaussian primitive, RaDe-GS (Zhang et al., 2024) proposes a method to rasterize depth and normal for a camera ray. Specifically, depth is calculated by finding the intersection point to a Gaussian primitive. Normal can then be calculated by finding a plane formed by the intersection point and the center of the Gaussian. Subsequently, depth $d$ and normal $n$ can simply be rasterized to pixel $\mathbf{u} = (u, v)$ as $d = z_c + p^\top \Delta \mathbf{u}_c$, $n = -\mathbf{J}^\top (q \quad 1)^\top$, where $z_c$ is the depth of the Gaussian center, $\Delta \mathbf{u}_c = (u_c - u, v_c - v)^\top$, $u_c$ and $v_c$ are Gaussian centers projected to screen space. Here, $q$ and $p$ are vectors that hold the first two elements of $\hat{q}$ and $\hat{p}$ respectively, which can be calculated using Gaussian center and covariance as

$$\hat{q} = \frac{\mathbf{v}'\boldsymbol{\Sigma}'^{-1}}{\mathbf{v}'^\top \boldsymbol{\Sigma}'^{-1}\mathbf{v}'}, \quad \hat{p} = \frac{z_c}{t_c}\hat{q}, \tag{3}$$

where $\mathbf{v}' = (0,0,1)^\top$, $z_c$ and $t_c$ are depth of Gaussian center in Euclidean space and ray space, respectively. Readers may refer to the original paper for detailed derivation.

In fact, 2DGS (Huang et al., 2024) and GOF (Yu et al., 2024b) propose similar methods of locating ray-Gaussian intersection, *where covariance is the key determinant of depth and normal rasterization,* as exemplified in Equation 3. However, an independantly trainable covariance can be oriented in any directions and thus can significantly differ from the distribution of adjacent Gaussians, as illustrated in Figure 3. As a result, rasterizing depth and normal from such covariance does not necessarily reflect local geometry.

## 3.2 COVARIANCE PARAMETERIZATION WITH ADJACENT GAUSSIANS

Instead of defining a fully trainable covariance matrix for every Gaussian primitive, our method defines the covariance as a function of adjacently located Gaussians, so that a Gaussian covariance can reflect the local geometry formed by its neighbors.

**Parameterization with Singular Value Decomposition**  For a given Gaussian primitive $\mathcal{G}$, consider the set of positions $P = [\boldsymbol{\mu}; \boldsymbol{\mu}^1; \cdots \boldsymbol{\mu}^K]$, where $\boldsymbol{\mu}^k$ is the mean of $\mathcal{G}^k$, the $k$th-nearest neighbor of $\mathcal{G}$. We first calculate the covariance of these points as

$$\Gamma = \text{Var}(P) = \frac{1}{K}(P - \bar{\boldsymbol{\mu}})(P - \bar{\boldsymbol{\mu}})^\top, \tag{4}$$

where $\bar{\boldsymbol{\mu}} = \frac{1}{K+1}(\boldsymbol{\mu} + \sum_{k \in K} \boldsymbol{\mu}^k)$. A naive approach is to simply use $\Gamma$ as a covariance of the Gaussian. However, we instead decompose $\Gamma$ to retrieve the orientation of local plane that these neighboring Gaussians form. Specifically, we decompose $\Gamma$ into principal components using Singular Value Decomposition (SVD) (Klema & Laub, 1980) as

$$\Gamma = U\Lambda V^\top, \tag{5}$$

where $V = [\mathbf{v}_1; \mathbf{v}_2; \mathbf{v}_3]$ are orthonormal eigenvectors and $\Lambda = diag(\lambda_1, \lambda_2, \lambda_3)$ are eigenvalues. These eigenvalues indicate the variance of the point distribution along each eigenvector ordered from largest to smallest, or $\lambda_1 \geq \lambda_2 \geq \lambda_3 \geq 0$. The first two eigenvectors lie on a local plane formed by the neighboring Gaussians, with the third eigenvector oriented perpendicular to this plane. Therefore, we can define the covariance matrix of a Gaussian using these eigenvectors and learnable scale $S$ as $\boldsymbol{\Sigma} = VSS^\top V^\top$. Gradients will be back-propagated to $P$ to update $\boldsymbol{\Sigma}$, so that the means of neighboring Gaussians are jointly optimized to form a desired $\boldsymbol{\Sigma}$.

However, defining the rotation factor of $\boldsymbol{\Sigma}$ solely from $K$ Gaussian neighbors may limit the expressiveness of the scene. To address this, we introduce a residual rotation term, $\Delta R$. This term is initialized as an identity matrix and is learned by integrating it into $V$, leading to the updated transformation $V' = \Delta RV$. Our Gaussian covariance is then computed as follows:

$$\boldsymbol{\Sigma} = V'SS^\top V'^\top. \tag{6}$$

It is crucial to regularize the residual rotation to preserve the inductive bias we have established. To achieve this, we introduce the following regularization term:

$$\mathcal{L}_r = |\Delta r - r_{\text{identity}}|, \tag{7}$$

where $\Delta r$ represents $\Delta R$ in quaternion form, and $r_{\text{identity}}$ is the identity quaternion. Note that we do not train $\Delta R$ during the earlier optimization stages, where most geometry is determined, to prioritize inductive bias over expressibility. Details and analysis on the residual rotation regularization can be found in Appendix A.3.

## 3.3 EIGENSYSTEM REGULARIZATION FOR LOCALLY SMOOTH PLANE ASSUMPTION

In this section, we introduce methods to regularize the eigensystem, the set of the eigenvectors paired with their eigenvalues calculated from Equation 5, in order to generate smooth and coherent local surfaces. We first regularize the smallest eigenvalue to align the neighboring Gaussians along a local plane. Furthermore, we enhance surface smoothness by assimilating the normals of adjacent planes proportional to the surface planarity.

### 3.3.1 EIGENVALUE REGULARIZATION

Since photometric loss by itself is geometrically ill-posed, there can be cases when a set of neighboring Gaussians do not form a plane (*i.e.* distributed uniformly toward all directions). Thus, positions

of these neighboring Gaussians need to be optimized toward a local plane to make the set of locally distributed Gaussians to behave as a surface.

Meanwhile, an eigenvalue reflects the amount of variance along its corresponding eigenvector. Thus, smaller $\lambda_3$ indicates that the Gaussians are distributed closer to the local plane whose normal is defined by the third eigenvector $\mathbf{v}_3$. To optimize our model as such, we simply minimize $\lambda_3$ using the following loss:

$$\mathcal{L}_{\text{eigval}} = \lambda_3. \tag{8}$$

### 3.3.2 EIGENVECTOR REGULARIZATION

In order to form locally smooth surfaces, we propose a regularization loss to assimilate the normals of neighboring local planes. Specifically, we minimize the cosine distance between the third eigenvectors of adjacent Gaussians using the following loss:

$$\mathcal{L}_{\text{eigvec}} = \frac{1}{K} \sum_{k \in K} \gamma^2 (1 - \mathbf{v}_3 \cdot \mathbf{v}_3^k), \tag{9}$$

where $\mathbf{v}_3^k$ is the third eigenvector of covariance calculated with $K$ nearest neighbors of the $k$-th neighboring Gaussian using Equation 5. Note that cosine distance is computed proportional to local planarity $\gamma$, which we elaborate in the next paragraph.

**Local Planarity Estimation with Eigenvalues**   When Gaussians express complex surface geometry, adjacent local planes do not necessarily share similar normal vectors. To quantify such cases, we define the local planarity formed by the neighboring Gaussians by leveraging the eigenvalues calculated from Sec. 3.2. Formally, $\gamma$ is defined as follows:

$$\gamma = 1 - \frac{3\lambda_3}{\sum_{j=\{1,2,3\}} \lambda_j}. \tag{10}$$

Note that $0 \leq \gamma \leq 1$ since eigenvalues are positive and descending. A value of $\gamma = 1$ indicates a completely planar surface, while lower values suggest otherwise. Also, $\gamma$ is detached from the optimization graph, as minimizing $\gamma$ can increase $\lambda_1$ and $\lambda_2$, which is irrelevant to accurate geometry reconstruction.

### 3.4 SPARSITY-AWARE ADAPTIVE DENSITY CONTROL

ADC densifies Gaussians with a high densification signal as in Equation 2. However, for homogeneously colored regions with low view-dependency (*i.e.*, extremely shadowed region as Figure 7), Gaussians yield low densification signals even when they are sparsely distributed, as a few number of points are enough to express the color of this relatively simple regions. However, such a case is problematic for accurate geometry reconstructions, as planes formed by these sparse neighbors are not local anymore, which may cause over-smoothing. We propose an additional densification algorithm to prevent such cases by selectively augmenting the number of Gaussians in regions with low densification signals and high sparsity. We term this strategy Sparsity-aware Adaptive Density Control (SADC) and use this along with the existing ADC (Kerbl et al., 2023).

Specifically, Gaussians with densification signal lower than $\tau$ are first selected. Then, we calculate the mean distance with its $K$-nearest Gaussians. The region these $K$ neighbors cover is considered sparse if the distance is greater than $\eta_{min} \cdot \tau_s$. We borrow the split threshold $\tau_s$ to measure sparsity (Gaussian is split into two if its scale is larger than $\tau_s$ (Kerbl et al., 2023)) as $\tau_s$ can be regarded as the maximal expectation of a Gaussian size. We also set the upper bound of sparsity $\eta_{max} \cdot \tau_s$ to reject background points. $\eta_{min} = 3.0$ and $\eta_{max} = 10.0$ are hyper-parameters.

Once a Gaussian is considered low-signaled and sparse, we interpolate $N$ additional Gaussians from each neighbor. Specifically, we sample the center of the interpolated Gaussian from a Gaussian distribution, where its mean is interpolated between the center of the original Gaussian and its neighbor and the covariance is inherited from the original Gaussian. All other properties, such as opacity $\alpha$ and SH color $\mathbf{c}$, are copied from the original Gaussian, except for $S$, which is scaled by $\frac{1}{N+1}$ to reflect the distance between adjacent Gaussians. The detailed algorithm is provided in Algorithm 1.

---

**Algorithm 1** Sparsity-aware Adaptive Density Control (SADC)

---

1:  $\mathcal{G}, \mathcal{G}' = \{\}, K, N$
2:  **if** $\frac{\partial L'}{\partial \mathbf{u}} < \tau$ **then**                                                                                      ▷ Low densification signal
3:      $d = \frac{1}{K} \sum_{k \in K} \left\| \boldsymbol{\mu} - \boldsymbol{\mu}^k \right\|_2$
4:      **if** $d > \eta_{\min} \cdot \tau_s$ and $d < \eta_{\max} \cdot \tau_s$ **then**                                                    ▷ High sparsity signal
5:          **for** $k \leftarrow 1$ to $K$ **do**
6:              **for** $n \leftarrow 1$ to $N$ **do**
7:                  $\boldsymbol{\mu}' \sim \mathcal{N}(\boldsymbol{\mu} + \frac{n}{N+1}(\boldsymbol{\mu}^k - \boldsymbol{\mu}), \frac{1}{N+1}\boldsymbol{\Sigma})$                       ▷ Interpolative sampling
8:                  $\boldsymbol{\alpha}' \leftarrow \boldsymbol{\alpha}, \mathbf{c}' \leftarrow \mathbf{c}, \mathbf{S}' \leftarrow \frac{1}{N+1}\mathbf{S}$
9:                  $\mathcal{G}'$.append($\{\boldsymbol{\mu}', \boldsymbol{\alpha}', \mathbf{c}', \mathbf{S}'\}$)
10: **return** $\mathcal{G}'$

---

### 3.5 Optimization

We supervise our model with RGB loss $\mathcal{L}_c = (1 - \beta)\mathcal{L}_1 + \beta\mathcal{L}_{D-SSIM}$ following Kerbl et al. (2023). In addition, we render depth and normal maps to screen space using rasterizer from Zhang et al. (2024), which are then supervised in screen space using depth-distortion and depth-normal consistency loss (Huang et al., 2024):

$$\mathcal{L}_d = \sum_{i,j} \omega_i \omega_j |z_i - z_j|, \quad \mathcal{L}_n = \sum_i \omega_i(1 - n_i^\top \tilde{n}), \tag{11}$$

where $\omega_i = \alpha_i \mathcal{G}_i^{2D}(\mathbf{u}) \prod_{j=1}^{i-1}(1 - \alpha_j \mathcal{G}_j^{2D}(\mathbf{u}))$ is the blending weight of the $i$-th intersection, $z_i$ is the depth of intersection points, $\tilde{n}$ is the normal map estimated from gradient of depth map, and $n_i$ is per-Gaussian normal rasterized with Equation 3. We empirically learned that employing $\mathcal{L}_d$, $\mathcal{L}_n$ along with $\mathcal{L}_{\text{eigvec}}$ and $\mathcal{L}_{\text{eigval}}$ yield the best results. We also discuss in Appendix. A.1 on distinct behaviors of $\mathcal{L}_n$ and $\mathcal{L}_{\text{eigvec}}$, which may look similar. Finally, our loss can be described as:

$$\mathcal{L} = \mathcal{L}_c + w_{\text{eigval}}\mathcal{L}_{\text{eigval}} + w_{\text{eigvec}}\mathcal{L}_{\text{eigvec}} + w_r\mathcal{L}_r + w_d\mathcal{L}_d + w_n\mathcal{L}_n, \tag{12}$$

where we set $w_{\text{eigval}} = 10$, $w_{\text{eigvec}} = 0.4$, $w_r = 0.01$. We follow GOF (Yu et al., 2024b) and RaDe-GS (Zhang et al., 2024) and set $w_d = 100$, and $w_n = 0.05$ for all experiments. We report detailed training strategy in Appendix A.2.

## 4 Experiment

We compare our methods with SuGaR (Guédon & Lepetit, 2024), 2DGS (Huang et al., 2024), GOF (Yu et al., 2024b), and RaDe-GS (Zhang et al., 2024), which are state-of-the-art surface reconstruction methods based on 3DGS (Kerbl et al., 2023). Implicit representation-based methods such as NeRF (Mildenhall et al., 2021), VolSDF (Yariv et al., 2021), NeuS (Wang et al.), and Neuralangelo (Li et al., 2023) are also compared.

**Mesh Extraction**    Depth maps are rendered from all training views to construct Truncated Signed Distance Fields (TSDF) (Curless & Levoy, 1996). From TSDF, Marching Cube (Lorensen & Cline, 1998) is used to construct a mesh.

**Datasets and Evaluation**    Since our objective is an accurate surface reconstruction, we experimented on DTU (Jensen et al., 2014) benchmark to compare Chamfer Distance (CD) on surface reconstructions. To measure the accuracy of surface normal as another aspect of geometry evaluation, we experimented on Synthetic NeRF (Mildenhall et al., 2021), which includes ground-truth normal maps. We measured Normal Similarity Score (NSS) between rendered normal and ground-truth normal maps as

$$\text{NSS} := \frac{1}{\|M\|} \sum_i M_i \cdot \frac{n_i^\top n_i^{\text{gt}}}{\|n_i\| \|n_i^{\text{gt}}\|}, \tag{13}$$

where $M$ is an optional GT mask, $n_i$ and $n_i^{\text{gt}}$ are rendered normal and GT normal at $i$-th pixel.

Table 1: **Quantitative comparison on the DTU Dataset**. We report CD and average optimization time on different methods. Best, second best, and third best within explicit/implicit methods are marked as red, orange, and yellow, respectively. ***Underlined italic bold*** denotes the best among all.

| | | 24 | 37 | 40 | 55 | 63 | 65 | 69 | 83 | 97 | 105 | 106 | 110 | 114 | 118 | 122 | Mean | Time |
|---|---|---|---|---|---|---|---|---|---|---|---|---|---|---|---|---|---|---|
| implicit | NeRF | 1.90 | 1.60 | 1.85 | 0.58 | 2.28 | 1.27 | 1.47 | 1.67 | 2.05 | 1.07 | 0.88 | 2.53 | 1.06 | 1.15 | 0.96 | 1.49 | >12h |
| | VolSDF | 1.14 | 1.26 | 0.81 | 0.49 | 1.25 | 0.70 | 0.72 | 1.29 | 1.18 | 0.70 | 0.66 | 1.08 | 0.42 | 0.61 | 0.55 | 0.86 | >12h |
| | NeuS | 1.00 | 1.37 | 0.93 | 0.43 | 1.10 | 0.65 | 0.57 | 1.48 | 1.09 | 0.83 | 0.52 | 1.20 | 0.35 | 0.49 | 0.54 | 0.84 | >12h |
| | Neuralangelo | *0.37* | 0.72 | 0.35 | *0.35* | 0.87 | *0.54* | *0.53* | 1.29 | *0.97* | 0.73 | *0.47* | *0.74* | *0.32* | *0.41* | *0.43* | *0.61* | >12h |
| explicit | 3DGS | 2.14 | 1.53 | 2.08 | 1.68 | 3.49 | 2.21 | 1.43 | 2.07 | 2.22 | 1.75 | 1.79 | 2.55 | 1.53 | 1.52 | 1.50 | 1.96 | 13.8m |
| | SuGaR | 1.47 | 1.33 | 1.13 | 0.61 | 2.25 | 1.71 | 1.15 | 1.63 | 1.62 | 1.07 | 0.79 | 2.45 | 0.98 | 0.88 | 0.79 | 1.33 | 82.1m |
| | 2DGS | 0.46 | 0.80 | *0.33* | 0.37 | 0.95 | 0.86 | 0.80 | 1.25 | 1.25 | 0.73 | 0.67 | 1.24 | 0.39 | 0.65 | 0.47 | 0.75 | 14.1m |
| | GOF | 0.45 | 0.76 | 0.37 | 0.37 | 0.95 | 0.84 | 0.74 | 1.18 | 1.30 | 0.70 | 0.81 | 0.80 | 0.40 | 0.73 | 0.48 | 0.72 | 71.0m |
| | RaDe-GS | 0.46 | 0.74 | 0.34 | 0.38 | 0.81 | 0.76 | 0.76 | 1.21 | 1.22 | 0.66 | 0.70 | 0.86 | 0.36 | 0.68 | 0.47 | 0.69 | 16.2m |
| | **IBGS** (*Ours*) | 0.49 | *0.68* | 0.37 | 0.38 | *0.78* | 0.73 | 0.73 | *1.12* | 1.24 | *0.61* | 0.61 | 0.90 | 0.37 | 0.65 | 0.46 | 0.67 | 91.2m |

## 4.1 COMPARISON TO STATE-OF-THE-ARTS

We compare CD on DTU, and report the results in Table 1. Our method achieves the best performance among 3DGS-based methods. Compared to Neuralangelo, our method takes faster training time, while outperforming on a few scenes within dataset. This means that our method is a more computationally efficient choice while having a potential to outperform on some scenarios. We also report qualitative comparisons on DTU in Figure 1 and Figure 4. As can be observed, our method is robust against broken surface geometry reconstruction. We also observed that our method successfully reconstruct surface with curvature such as round foot instep of the doll, as observed in second row result in Fig 4.

We also compare NSS on NeRF-synthetic and report the results in Table 2. We report the complete list of NSS for each scene in Appendix A.6. Our method achieves the best NSS among SOTA 3DGS-based methods. Such outperformance can also be observed from qualitative comparison on surface reconstructions in Figure 6. Our method yields the most complete and smooth surface reconstructions.

Table 2: **Quantitative comparison on Synthetic NeRF dataset.** Our method yields the best Normal Similarity Score (NSS) among state-of-the-art 3DGS-based methods.

| | NSS ↑ | PSNR↑ | SSIM↑ | LPIPS↓ |
|---|---|---|---|---|
| 2DGS | 0.695 | 33.65 | 0.969 | 0.0317 |
| GOF | 0.697 | 33.71 | 0.969 | 0.0308 |
| RaDe-GS | 0.700 | 33.68 | 0.970 | 0.0304 |
| **IBGS** (*Ours*) | 0.703 | 33.70 | 0.971 | 0.0321 |

Although the Mip-NeRF360 dataset does not include a quantitative evaluation of geometry, we have provided a qualitative comparison of surface reconstruction and normal rendering in Fig.5. In particular, the surface reconstruction results displayed in Fig.5-(a) highlight the ability of our method to produce a highly accurate and plausible surface reconstruction. Another notable observation is the preservation of complex geometrical features, such as the intricate and bumpy texture of cloth surfaces, which remains intact under our regularization. Our method can thus be optimized toward smooth surfaces without compromising

Table 3: **Quantitative comparison on Mip-NeRF 360 Dataset.** Our method yields comparable novel view synthesis result especially on indoor scenes.

| | Outdoor Scene | | | Indoor scene | | |
|---|---|---|---|---|---|---|
| | PSNR ↑ | SSIM ↑ | LPIPS ↓ | PSNR ↑ | SSIM ↑ | LPIPS ↓ |
| NeRF | 21.46 | 0.458 | 0.515 | 26.84 | 0.790 | 0.370 |
| Deep Blending | 21.54 | 0.524 | 0.364 | 26.40 | 0.844 | 0.261 |
| Instant NGP | 22.90 | 0.566 | 0.371 | 29.15 | 0.880 | 0.216 |
| MERF | 23.19 | 0.616 | 0.343 | 27.80 | 0.855 | 0.271 |
| MipNeRF360 | 24.47 | 0.691 | 0.283 | 31.72 | 0.917 | 0.180 |
| 3D GS | 24.54 | 0.731 | 0.234 | 30.41 | 0.920 | 0.189 |
| Mip-Splatting | 24.65 | 0.729 | 0.245 | 30.90 | 0.921 | 0.194 |
| 2D GS | 24.34 | 0.717 | 0.246 | 30.40 | 0.916 | 0.195 |
| GOF | 24.82 | 0.750 | 0.202 | 30.79 | 0.924 | 0.184 |
| RaDe-GS | 25.17 | 0.764 | 0.199 | 30.74 | 0.928 | 0.165 |
| IBGS (*Ours*) | 24.64 | 0.723 | 0.213 | 30.69 | 0.932 | 0.161 |

high-fidelity reconstructions of complex surfaces. Such balance is crucial for generating high-quality surface representations across various geometrical complexities.

Our method demonstrates competitive performance in novel view synthesis on the MipNeRF-360 dataset, particularly for indoor scenes, as shown in Table 3. However, as observed from Figure 5-(b), PSNR does not reflect the quality of the underlying geometry. While GOF and RaDe-GS produce visually appealing RGB renderings on the floor, the underlying geometries observed from the normal renderings indicate inaccurate geometric structure with holes. In contrast, our approach delivers comparable visual quality and maintains a more accurate and consistent geometric structure. Such distinction between RGB and geometry rendering underscores the importance of evaluating visual fidelity and geometric accuracy when assessing the model as a scene reconstruction method.

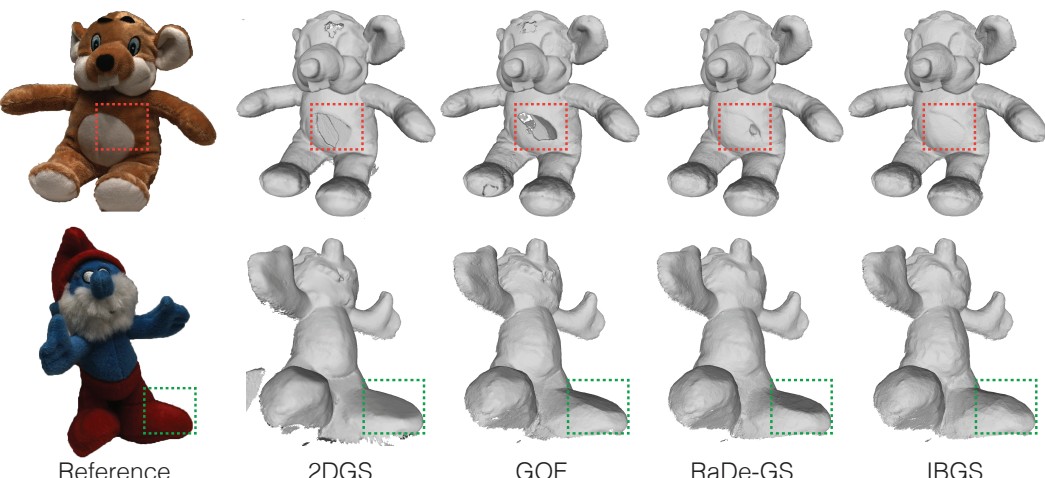

Figure 4: **Qualitative comparisons on DTU (Jensen et al., 2014) dataset.** IBGS yields the most plausible results among 3DGS-based methods.

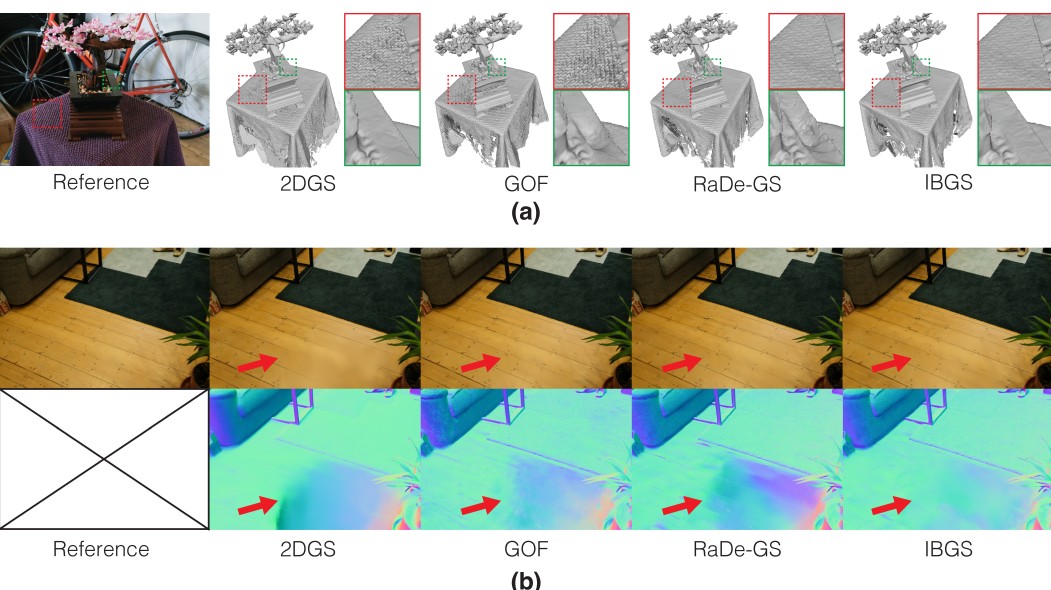

Figure 5: **Qualitative comparisons on MipNeRF360 (Barron et al., 2022) dataset** over **(a)** surface reconstruction, **(b)** novel view synthesis *(top)* and rendered normal maps *(bottom)*.

### 4.2 ABLATION STUDY

We conducted an ablation study for our proposed methods and reported the results in Table 4. Specifically, we ablate over *param.*, $\mathcal{L}_{\text{eigval}}$, $\mathcal{L}_{\text{eigvec}}$, where *param.* refers to our covariance parameterization with eigenvectors of neighboring Gaussians. Notably, full regularization without our parameterization yields the worst result. For regularizations, we can empirically conclude that the absence of either $\mathcal{L}_{\text{eigval}}$ or $\mathcal{L}_{\text{eigvec}}$ results in a bad result. Such observation corresponds to the design philosophy of our method, as eigenvector assimilation is significant only when it can behave as local plane normal, which can be achieved by minimizing the third eigenvalue.

Furthermore, SADC exhibits improvements in both quantitative and qualitative outcomes from Table 4 and Figure 7 respectively. When SADC is not applied, certain regions, particularly the shadowed area between the arms, exhibit excessive smoothing, leading to the loss of geometric details. In contrast, SADC effectively addresses this issue by ameliorating the over-smoothed surfaces. Such observation highlights the role of SADC in successfully densifying the sparse regions suitable for our inductive bias to assume planarity in local space.

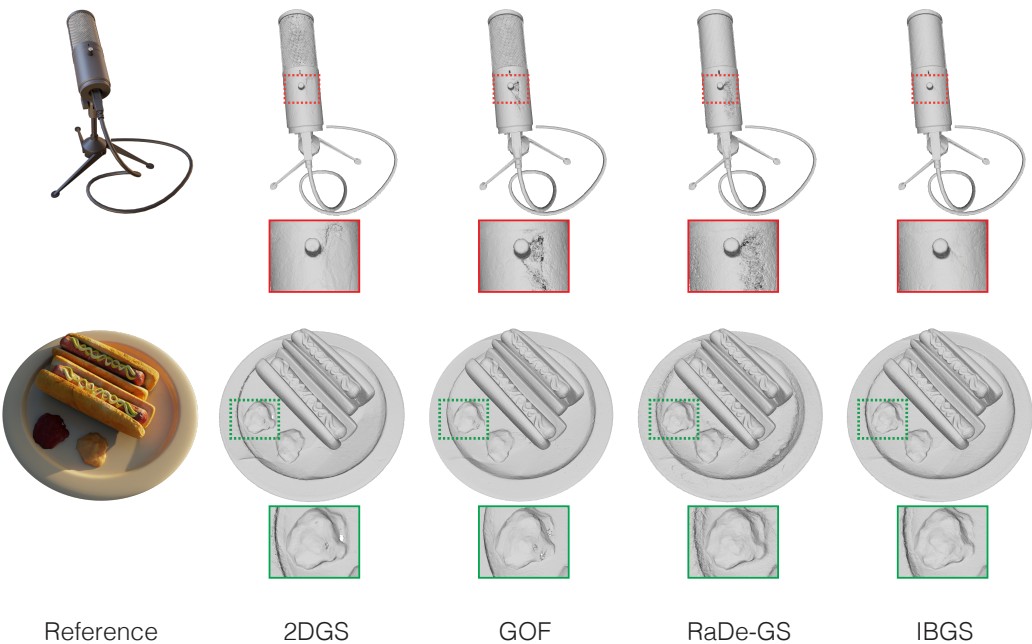

Reference      2DGS      GOF      RaDe-GS      IBGS

Figure 6: **Qualitative comparisons on NeRF-Synthetic (Mildenhall et al., 2021) dataset**. IBGS yields the most high-quality surface reconstruction results.

|  | CD $\downarrow$ |
|---|---|
| Ours *w.o/ param.* | 0.702 |
| Ours *w.o/* $\mathcal{L}_{\text{eigval}}$ | 0.695 |
| Ours *w.o/* $\mathcal{L}_{\text{eigvec}}$ | 0.692 |
| Ours *w.o/* SADC | 0.682 |
| Ours | 0.674 |

Table 4: **Ablation study** of our proposed methods on DTU. Our parameterization method significantly improves the performance, followed by the regularization techniques and SADC.

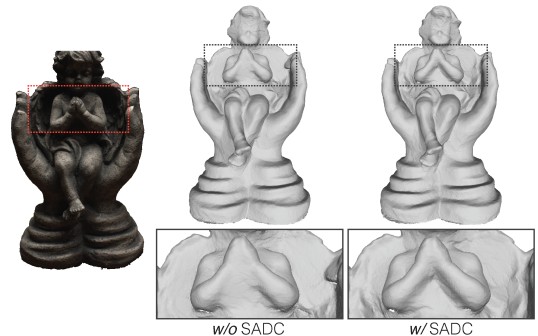

*w/o* SADC      *w/* SADC

Figure 7: **Qualitative ablation results on SADC**. Without SADC, unilluminated regions tend to be over-smoothed.

## 5   CONCLUSION

In this work, we presented IBGS, a novel 3D Gaussian model incorporating geometrically accurate parameterization and regularization. Recognizing the critical role of covariance in determining surface normals, we argued that it should be parameterized based on the distribution of neighboring Gaussians for precise geometric representation rather than being entirely trainable. To achieve this, we applied SVD to derive the principal axes and corresponding variances of neighboring Gaussians, or the eigenvectors and eigenvalues. The model's covariance is then defined by combining these eigenvectors with a learnable scaling matrix. Additionally, we introduced regularization techniques that minimize the third eigenvalue, encouraging neighboring Gaussians to align on a local plane and align the third eigenvectors, which approximate local plane normals, for smoother surface representations. We also proposed SADC to densify regions with low densification signal. Our experiments on state-of-the-art benchmarks demonstrate that IBGS provides the most geometrically accurate representation among 3D Gaussian-based methods while requiring significantly less training time than implicit neural representations.

## REPRODUCIBILITY STATEMENT

To ensure the reproducibility of our method, we provided all the hyperparametes we used in Section 3.4 and Section 3.5. We also provide experimental details in Section 4, pseudocode on SADC in Algorithm 1 and details on training schedules in Appendix A.2.

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

# A APPENDIX

## A.1 COMPARING $\mathcal{L}_n$ WITH $\mathcal{L}_{\text{EIGVEC}}$

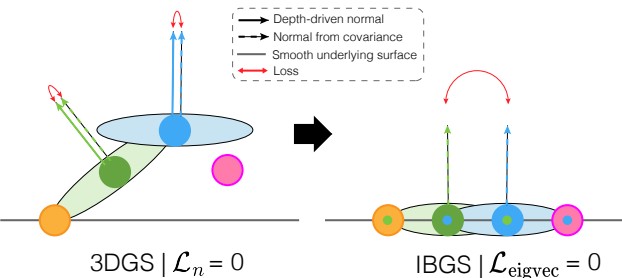

Figure 8: Illustrations on *(Left)* a case when vanilla 3DGS satisfy $\mathcal{L}_n$ yet cannot converge toward the underlying smooth surface, and *(Right)* how $\mathcal{L}_{\text{eigvec}}$ induce local planes to reside on a single smooth surface by having to adjust neighboring Gaussians ($K = 2$) to assimilate the third covariance axis.

In this section, we would like to clarify a potential confusion on the working mechanism between $\mathcal{L}_{\text{eigvec}}$ under our parameterization versus depth-normal consistency loss $\mathcal{L}_n$, a geometry regularization method in screen space.

Primarily, $\mathcal{L}_n$ is NOT EXPLICITLY designed for smooth surfaces, but is designed to ***make an agreement between rendered normal and depth***. In other words, if one or all of depth/normal maps are not smooth, none of the rendered normal and depth will likely be optimized to form a smooth surface. For example, consider the illustration in Figure 8, where green/blue Gaussians are our optimization targets, and orange/pink Gaussians are other adjacent Gaussians. Figure 8 shows an example of vanilla 3DGS optimized with $\mathcal{L}_n$. In this example, normal and depth-driven normal completely agree with each other (depth-driven normal is calculated with a plane formed by two closest Gaussians), but this does not guarantee the Gaussians to be converged toward a smooth underlying surface, yielding non-planar distribution of adjacent Gaussians.

On the other hand, $\mathcal{L}_{\text{eigvec}}$ assimilates the neighboring third covariance axes (blue and green dotted arrows in the right-side figure of Fig 8), which requires back-propagation of gradients to adjacent Gaussians that form these covariances. Aligning covariances then require the neighboring Gaussians to be aligned correspondingly such that the third covariance axes become parallel. As a result, the neighboring Gaussians are forced to form a locally smooth plane under our inductive bias. Due to such working mechanism, $\mathcal{L}_{\text{eigvec}}$ works only under our parameterization, as was empirically validated in Table 4, where $\mathcal{L}_{\text{eigvec}}$ and $\mathcal{L}_{\text{eigval}}$ without our inductive bias performs worse.

Also, note that $\mathcal{L}_{\text{eigvec}}$ is conducted in 3D space whereas $\mathcal{L}_n$ regularizes rendered maps in screen space. Thus, they are not strictly comparable.

## A.2 DETAILED TRAINING STRATEGY

Given total optimization steps $I$, we divide the process into multiple phases, which can be summarized in Equation 14. We learned that imposing $\mathcal{L}_{\text{eigval}}$ and $\mathcal{L}_{\text{eigvec}}$ from the start brings unstable training. Thus, we restore coarse geometry based on our inductive bias only, and apply our regularization term afterwards. Also, when depth distortion and depth normal consistency loss is applied, we disable our parameterization by initializing a trainable rotation matrix using the rotation matrix determined at step $\frac{I}{2}$.

$$\mathcal{L} = \begin{cases} \mathcal{L}_c & i \leq \frac{I}{6} \\ \mathcal{L}_c + \beta_{\text{eigval}}\mathcal{L}_{\text{eigval}} + \beta_{\text{eigvec}}\mathcal{L}_{\text{eigvec}} & \frac{I}{6} < i \leq \frac{I}{3} \\ \mathcal{L}_c + \beta_{reg}\mathcal{L}_r & \frac{I}{3} < i \leq \frac{I}{2} \\ \mathcal{L}_c + \beta_d\mathcal{L}_d + \beta_n\mathcal{L}_n & \frac{I}{2} < i \leq I \end{cases} \tag{14}$$

### A.3 Bag of Regularization Techniques for Residual Rotation

It is important to keep the balance between our parameterization and expressibility when introducing residual rotation, $\Delta R$. To prioritize inductive bias, $V$ is used instead of $V'$ until step $\frac{I}{3}$ given a total iteration step $I$ in order to force our inductive bias in earlier stages of training, where we empirically learned that most geometry is determined. Then, we introduce $\Delta R$ from $\frac{I}{3}$ along with the proposed regularization term for residual rotation, $\mathcal{L}_r$. We may refer to these as a bag of regularization techniques for residual rotation (BRR).

To demonstrate that BRR leads to faithful reflection of inductive bias, we visualize the angles of residual rotation when trained with and without BRR in Figure 9, where without BRR, rotation residual is trained throughout all iterations without $\mathcal{L}_r$. As can be observed, most of the residual rotations is very close to $0°$ under our regularization, meaning that IBGS reasonably follows the inductive bias our parameterization forms. Also note that without BRR, chamfer distance results in $0.692$ in DTU benchmark, where with BRR result in $0.674$. This also implies that sticking to our parameterization leads to more accurate geometry.

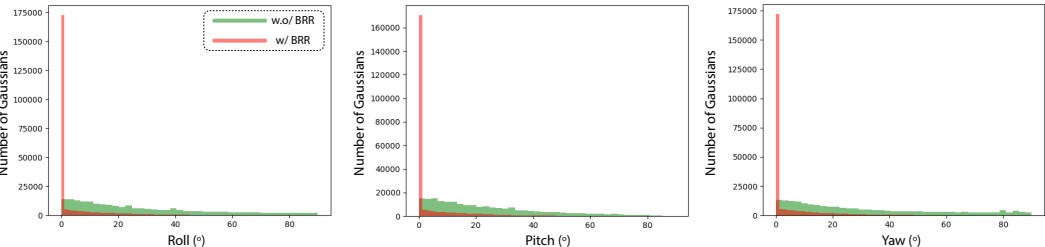

Figure 9: Angles of residual rotation histograms with and without BRR. BRR effectively regularizes residual rotation, which means that IBGS well retains the inductive bias formed with our parameterization with eigenvectors.

### A.4 K-Nearest Neighbor Calculation

We used Faiss-GPU (Johnson et al., 2019) library for fast computation of $K$ nearest neighbors. Ideally, one may calculate the kNN graph for every iteration, which also causes longer training time. Instead, we update KNN graph for every 100-th iteration to balance between computational efficiency and accuracy.

### A.5 Ablation on $K$

In this section, we present the ablation study results for the parameter $K$ on the DTU Dataset, which were conducted to determine the optimal value of $K$ for achieving the best performance. Based on the experimental findings, we observed that setting $K = 5$ produced the most favorable outcomes in terms of the Chamfer Distance (CD).

| $K$ | 2 | 3 | 4 | 5 | 6 | 7 | 8 | 9 | 10 |
|---|---|---|---|---|---|---|---|---|---|
| CD | 0.701 | 0.709 | 0.684 | **0.674** | 0.678 | 0.686 | 0.684 | 0.688 | 0.692 |

Table 5: Ablation results of $K$ on DTU Dtaset Jensen et al. (2014)

### A.6 Additional Qualitative and Quantitative Results

We report additional qualitative results on DTU and NeRF-synthetic dataset in Figure 10 and Figure 11, respectively. The results demonstrate high-fidelity surface reconstructions. It is also notable that our parameterization and geometry regularization does not disturb expressing fine geometric details.

### A.7 Application of our method to other 3DGS-based methods

In addition, we also experimented how our proposed method can improve surface reconstruction quality with other 3DGS-based methods. For that, we have applied our method on 2DGS and GOF,

| | Chair | Drums | Ficus | Hotdog | Lego | Materials | Mic | Ship | Mean |
|---|---|---|---|---|---|---|---|---|---|
| 2DGS Huang et al. (2024) | 0.670 | 0.667 | 0.643 | 0.795 | 0.685 | 0.724 | 0.612 | 0.765 | 0.695 |
| GOF Yu et al. (2024b) | 0.673 | 0.666 | 0.653 | 0.793 | 0.691 | 0.722 | 0.611 | 0.769 | 0.697 |
| RaDe-GS Zhang et al. (2024) | 0.686 | 0.662 | 0.658 | 0.792 | 0.679 | 0.738 | 0.609 | 0.778 | 0.700 |
| IBGS *(Ours)* | 0.689 | 0.670 | 0.659 | 0.797 | 0.686 | 0.734 | 0.613 | 0.778 | 0.703 |

Table 6: Normal Similarity Score (**NSS**) on all scenes of NeRF-Synthetic Mildenhall et al. (2021)

| | 24 | 37 | 40 | 55 | 63 | 65 | 69 | 83 | 97 | 105 | 106 | 110 | 114 | 118 | 122 | Mean |
|---|---|---|---|---|---|---|---|---|---|---|---|---|---|---|---|---|
| 2DGS Huang et al. (2024) + *Ours* | 0.48 | 0.78 | 0.34 | 0.37 | 0.92 | 0.81 | 0.78 | 1.21 | 1.25 | 0.68 | 0.62 | 1.10 | 0.39 | 0.66 | 0.47 | 0.72 |
| GOF Yu et al. (2024b) + *Ours* | 0.45 | 0.73 | 0.36 | 0.37 | 0.88 | 0.79 | 0.73 | 1.14 | 1.31 | 0.62 | 0.74 | 0.81 | 0.41 | 0.73 | 0.48 | 0.70 |

Table 7: Quantitative results on applying our method using rendering equation from 2DGS and GOF

and experimented on DTU Dataset. As reported in Table. 7, our method brings reasonable improvements regardless of the rendering equation.

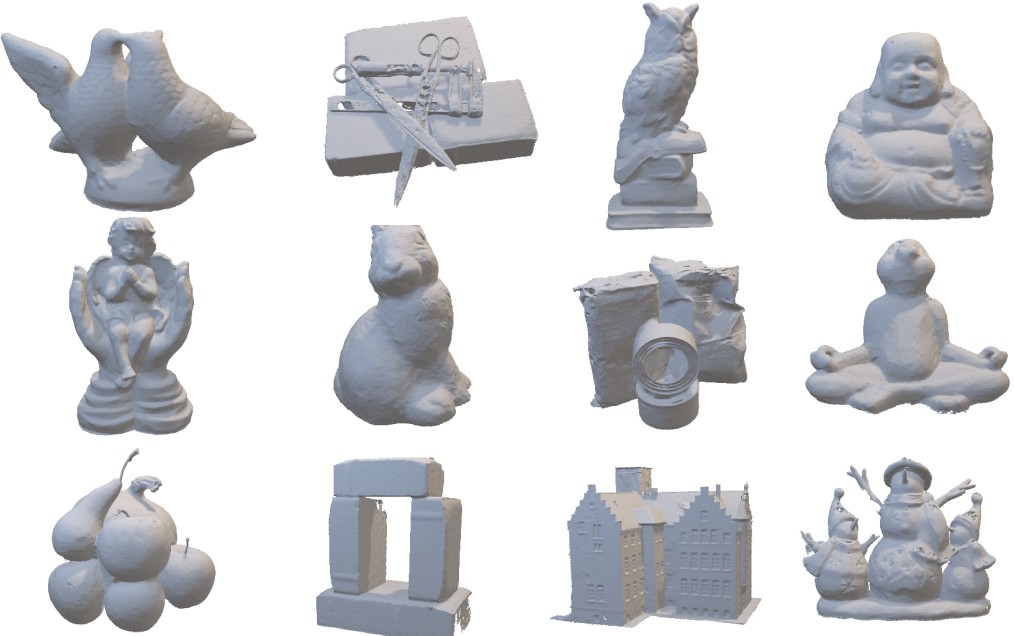

Figure 10: Extensive surface reconstruction results of IBGS on DTU (Jensen et al., 2014)

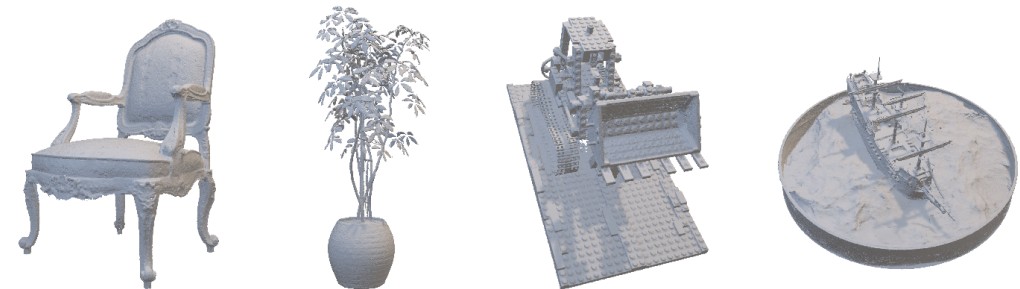

Figure 11: Extensive surface reconstruction results of IBGS on NeRF Synthetic (Mildenhall et al., 2021) Dataset.