# OpenReview forum: "Injecting Inductive Bias to 3D Gaussian Splatting for Geometrically Accurate Radiance Fields"
_ICLR.cc/2025/Conference — Submitted to ICLR 2025_

### Official Review · Reviewer_2Feu · 2024-10-24

**Soundness:** 3
**Presentation:** 3
**Contribution:** 2
**Rating:** 3
**Confidence:** 5

**Summary:**

This paper aims to overcome the limitations of 3DGS in geometry reconstruction, with a focus on regularization techniques. Previous methods independently define each Gaussian novel vector and only used screen space regularization. However, the paper argues that the normal vector of a Gaussian should depend on its local neighbors. To incorporate structural priors (also referred to as inductive bias in the paper), the paper proposes to parameterize the covariance of each primitive based on its neighborhood. Several regularizations based on the parameterized covariance are then integrated. Experiments suggest that this approach achieves state-of-the-art geometry reconstruction results across multiple datasets.

**Strengths:**

1. The paper is well-written and easy to follow. The results seem reproducible given sufficient details.

2. Using local neighbors to compute the orientation seems original and reasonable for achieving better results. The visualization also consistently suggests that the proposed regularizations alleviate some holes presented in previous methods.

**Weaknesses:**

1. The significance of the proposed method appears to be limited as it is built upon RaDeGS with additional regularization. However, it only results in a 0.02 gain in the DTU dataset while being five times slower than RaDeGS.

2. I understand that the normal vector should ideally be derived from the density distribution of the superposition of Gaussian functions.  However, However, I believe that the current method doesn't address the core issue. From a geometric view, the normal vector should not depend on the viewing angle. In Section 3.1, the normal is calculated by determining a plane formed by the intersection point and the center of the Gaussian, which is view-dependent. Consequently, the definition of the normal may contradict the argument based on the density distribution and the orientation defined by the covariance obtained through local neighbors.

3. Lastly, constructing local covariance using KNN may pose challenges. Each Gaussian represents a three-dimensional signal, not a zero-dimensional point, so we should consider its anisotropic nature when taking neighbor information into account. For example, a 3D Gaussian with a density that encompasses a point but with its mean located away from that point.

**Questions:**

1. What is the cause of lower training efficiency? Is it because of the regularization? What is the number of primitives?

2. How can you address the problem mentioned in the weakness or if the impact is negligible in practice?

---

> ### Author Response · Authors · 2024-11-19
>
> **We thank the reviewer for proposing novel perspectives, feedbacks and potential improvements of our work.**
>
> >
> > ## Shouldn't the normal vector be ideally derived from superposition of Gaussians?
> >
>
> Thank you for asking the interesting question. Before answering to the reviewer's question, we would like to revisit the Splatting algorithm **[E]**, the core rasterization method for most 3DGS-based frameworks including ours. Given a set of points, each point is assigned with a reconstruction kernel $r(x)$, which is usually represented with a Gaussian function. The kernel represents the weight of impact the point has to its neighboring space in 3D. For rasterization, each point and its follwing reconstruction kernel is projectively transformed to screen-space in order to calculate the weight the point has on each pixel within the bounding box of the kernel. The key here is the affine approximation of the projective transformation for individual projection of Gaussian reconstruction kernel, followed by the volume rendering of these individually projected Gaussians.
>
> Meanwhile, what the reviewer seems to suggest is a continuous field in 3D represented with Gaussian Mixture Model (GMM). While this is an interesting approach, the current splatting algorithm is not designed to handle GMM. Thus, new algorithm may be proposed to somehow define and rasterize color, depth, and normal from a GMM model and rasterize them to screen-space, which we believe can be novel and interesting field of study.
>
> >
> > ## From a geometric view, the normal vector should not depend on the viewing angle? If it does, isn't there a problem that the third eigenvector and rendered normal do not match?
> >
>
> We sincerely thank the reviewer for the insightful question. First, we would like to clarify that a Gaussian is a function that covers a continuous space, NOT a zero-dimensional point as the reviewer kindly pointed out. The reviewer is correct if we are talking about the normal of a point.
> However, a Gaussian function is desinged to define different values for different locations around the Gaussian center, while the camera ray can be projected to any location around that space. Thus, any property including normal must be queried given a specific location around the Gaussian mean. In other words, it is better NOT to represent a single normal value for a single Gaussian. In fact, this is the philosophy of the more recent outperforming methods such as GoF and RaDe-GS compared to 2DGS, which rather assings a single normal vector per 2D Gaussian, or surfel.
>
> Due to these reasons, we did not overstate the third eigenvector $v_3$, as the normal vector of a Gaussian as a whole. Instead, we made a soft yet reliable link between the covariance and the rendered normal using Eq.(3), which RaDe-GS proposed and proved to work in the state-of-the-art performance. However, it would be an interesting research topic to make a strict and mathematical analysis between the third axis of covariance and the rendered normal given a camera ray under the rendering equation proposed by Eq.(3).
>
> However, we believe that our paper made a strong empirical evidence that Eq.(3) makes geometrically plausible relationship between the orientation of covariance and rendered normal given a camera ray.
>
> >
> > ## Shouldn't we take the continuous and anisotropic nature of adjacent Gaussians into account?
> >
>
> We agree with the reviewer that explicitly reflcting neighboring covariances can be another interesting research direction. Yet, we also believe that IBGS is partially working that way already. Specifically, the neighbor of one Gaussian can also be the neighbor of another Gaussian, meaning that the shared neighbor affects the covariance orientation of both Gaussians. In other words, updating one covariance affect the other covariance via their shared neighbors.
>
> >
> > ## CD improvement of 0.02 is too small?
> >
>
> Please kindly refer to the answer `1. On seemingly negligible improvement in quantitative geometry measures.` from the general response!
>
> >
> >## Training is too slow?
> >
>
> Please kindly refer to the answer `3. Analysis and discussion on training time` from the general response!
>
> >
> > ***Reference***
> >
>
> **[E]** Zwicker et al. "EWA volume splatting." Proceedings Visualization, IEEE, 2001.

---

> ### Comment · Reviewer_2Feu · 2024-11-19
> **The empirical study should address technical concerns and requires more experiments to assess its effectiveness.**
>
> I appreciate the efforts made by the authors; however, my questions and concerns have not been fully addressed. I still have reservations about the technique and its underlying theorem. The authors suggest that **"the properties of Gaussians should be defined with neighbors,"** but I am more interested in exploring **"how to define it theoretically and correctly?"** Below are my specific points:
>
> 1. Regarding view-dependent normals, the authors state that **"it is better NOT to represent a single normal value for a single Gaussian,"** which I agree with. However, my question is: **"Why is the normal of a Gaussian made view-dependent?"** Even in a case with varying normals, such as a sphere with spatially varying normals, the normals remain consistent and do not change with the viewer's perspective. The authors reference "GoF, RaDe-GS, 2DGS" to justify the view-dependent normal, but I do not believe these are gold standards to follow, given their known problems.
>
> 2. Concerning the superposition of primitives, I assume we all acknowledge that a ray-traced method for these Gaussians should yield the universally correct normal, based on the argument that "the normal should be the negative gradient of the density distribution." I am curious why the authors opted for a splatting approach. The only reason I can think of for this choice is speed. **However, your current algorithm does not seem to be significantly faster than a ray-traced Gaussian**. We all know that splatting (the forward mapping version of alpha blending) is an approximation of ray tracing (the backward mapping version of alpha blending). There is a clear mathematical relationship in this approximation, as seen in EWA Volume Splatting [1] (you did read this). **Can you also provide a mathematical derivation that demonstrates how your definition of the normal vector serves as a good approximation?**
>
> 3. Regarding the anisotropic nature of Gaussians, the method proposes estimating the normal from the mean of the Gaussians, which is analogous to estimating a normal vector from a point cloud. This concept is not new, as estimating normal vectors from point clouds is a well-established problem integrated into many tools like Open3D and MeshLab. My concern is that you cannot treat 3D Gaussians as a simple point cloud because Gaussians have shape, which may result in tools like KNN failing. For instance, a large Gaussian positioned far from the center may be overlooked during KNN processing. I personally prefer techniques that do not depend on KNN, as they can be sensitive to the choice of K and lead to inefficiencies during training, particularly when Gaussian points are updated frequently and KNN must be recalibrated.
>
> 4. It is also puzzling that the method claims "injecting such inductive bias" would be useful for 3D Gaussians, yet it still relies on screen-space regularizations from 2DGS.
>
> 5. Lastly, the contributions need refinement. The third contribution is not a true contribution, as it builds upon RaDeGS for surface reconstruction but renders it slower. In other words, the speed contribution is already made by what you follow, and you make it slower!!! It is also not state-of-the-art as it did not compare to some work like PGSR (https://arxiv.org/abs/2406.06521).
> > Our method achieves state-of-the-art performance in surface reconstruction tasks among 3DGS based methods while maintaining faster training time compared to implicit neural representation based methods
>
> **In summary, I believe that the current proposed method contains several apparent flaws in its current form**. It resembles an empirical study but lacks the necessary theoretical analysis. While I believe that empirical studies can benefit the community, this one lacks sufficient validation. I am open to the idea of an empirical study, but to fully validate the ideas and claims presented, I recommend that the authors include the following experiments (which I believe are quite easy to implement):
>
> 1. Benchmark the proposed method on Tanks and Temple. The authors have only conducted experiments on small-scale datasets like DTU. It remains unclear how the method performs on larger-scale datasets. Previous methods (like 2DGS, GoF, RaDe-GS, and VCR-Gaus) have all been benchmarked on this dataset, so it is surprising that this paper has not done so, especially considering it should be easy to implement.
>
> 2. Exclude the regularizations from previous methods. From the title and introduction, it seems that the proposed method should improve upon naive 3D Gaussians, so why does it still rely on screen space regularizations? There is no analysis provided to explain this dependence. I recommend that the authors evaluate the proposed method without $L_d$ and $L_n$.

---

> > ### Author Response · Authors · 2024-11-22
> > **Regarding view-dependent normals**
> >
> > >
> > > ## Regarding view-dependent normals
> > >
> >
> > We would like to answer view-depndency in wider perspective based on the reviewer's initial query on *"the definition of the normal may contradict the argument based on the density distribution and the orientation defined by the covariance obtained through local neighbors."*, which the reviewer concerned may exist due to view dependancy.
> >
> > First of all, we want to remind our analysis above on how the reviewer's suggestion on *"the definition of normal"*, or negative gradient of field (Gaussian in our case), is irrelevant for defining normal. We also showed that given that we are dealing with a **Gaussian function** as a primitive for density function, there cannot exist a golden standard or a strictly correct way on defining normal.
> >
> > **Given the above, the best we could do to prove our parameterization and regularization is to `experiment` our method using the existing `formulations` of rasterization.** Initially, we thought that 2DGS will get along with our method the best, because it is NOT view-dependant and explicitly makes the third eigenvector as a single normal vector for a Gaussian, which does not make the contradiction that the reviewer is concerned of. However, experimenting our method using 2DGS brought improvemnt in CD from `0.75` to `0.72`. Here, our method did bring improvements, but it was still worse than experimenting our method with RaDe-GS rasterizer.
> >
> >  **We believe that improvements in surface reconstruction using RaDe-GS rasterizer along with our method is the strong empirical proof that our eigenvector parameterization reasonably goes along with the soft link, (*which we admit is not strictly interpretable at this moment*), that RaDe-GS rasterization has formed between the covariance axis and the followingly rendered normal.**

---

> > ### Author Response · Authors · 2024-11-22
> > **Further empirical studies**
> >
> > >
> > > ### Screen-space regularization, $\mathcal{L}\_{d}$ and $\mathcal{L}\_{n}$, are applied to all competititve 3DGS-based methods for the experiments.
> > >
> >
> > We apologize for the confusion. Please note that all 3DGS-based methods except SuGaR (thus 2DGS, GOF, RaDe-GS and ours) employ the exact same screen-space regularization proposed by 2DGS, which is $\mathcal{L}\_{d}$ and $\mathcal{L}\_{n}$. Also, we used the same values for hyperparameters $\lambda\_{d}$ and $\lambda\_{n}$ for all methods. **Thus, difference between RaDe-GS and our method is caused purely by our proposed method only.**
> >
> > Following the reviewer's recommendation, we report experimental results on our method without $\mathcal{L}\_{d}$ and $\mathcal{L}\_{n}$.
> >
> > | RaDe-GS $- (\mathcal{L}\_{d}$ + $\mathcal{L}\_{n})$ | RaDe-GS | IBGS  $- (\mathcal{L}\_{d}$ + $\mathcal{L}\_{n})$ | IBGS |
> > |:--:|:--:|:--:|:--:|
> > |1.31|0.69|1.28|0.67|
> >
> > >
> > > ### Experiments on TnT dataset.
> > >
> >
> > Following the reviewer's recommendation, we report experimental results on TnT with F1 score, the common evaluation metrics for TnT. The results show that our method barely made improvements in outdoor scenes. We conjecture that our parameterization losses its assumption on large unbounded scenes, where Gaussians located far from camera has to be sparser, which is caused by the nature of inverse rendering and perspective projection. We make a further discussion on this limiation in `2. Failure case & limitations` from the general response.
> >
> > **Please note that ours instead have clear advantage in object-level surface reconstruction, as we additionally demonstrated with Normal Similarity Score (NSS) on NeRF-Synthetic dataset.**
> >
> >
> > | | GOF | RaDe-GS | IBGS (Ours) |
> > |:--:| :--:| :--:| :--:|
> > |Barn        |0.51 |0.43 | 0.41 |
> > |Caterpillar |0.40 |0.32 | 0.34 |
> > |Courthouse  |0.29 |0.21 | 0.23 |
> > |Ignatius    |0.67 |0.69 | 0.68 |
> > |Meetingroom |0.29 |0.25 | 0.26 |
> > |Barn        |0.59 |0.51 | 0.50 |
> > |Mean        |0.46 |0.40 | 0.40 |

---

> ### Author Response · Authors · 2024-11-22
> **Regarding the superposition of primitives (1/2)**
>
> **We sincerely thank the reviewer for the thorough review. Here are the answers to the questions.**
>
> Since there are many sections of answers, we would like to summarize the sections for easier understanding.
> 1. Regarding the superposition of primitives
> 2. Regarding view-dependent normals
> 3. Regarding the anisotropic nature of Gaussians and kNN
> 4. Further empirical studies.
>
> ---
>
> >
> > ## Regarding the superposition of primitives
> >
>
> Thank you for clarifying the question. Below, we compare our work with the suggested ideal framework where a scene is represented as functional fields with superposition of Gaussians, and normal is rendered by ray-tracing on that space, where each normal on the sampled point of a ray is calculated by negative gradient of a Gaussian function.
>
> ### **0. Formulating superpositioned Gaussian field using superpositioned Gaussain Field**
>
> We start off by defining how the procedure of ray-tracing based rendering of normal on superpositioned Gaussian field would look like. We will refer the superpositioned Gaussian function as Gaussian Mixture Model $p(x)$, which can be formulated as
>
> $$p(x) =  \log \left( \sum_{m=1}^M \pi\_m \mathcal{N}(x \mid \mu\_m, \Sigma\_m) \right),$$
>
> where $\pi\_m$ is summation weight and log is put for numerical stability. Then, the negative gradient of GMM with respect to $x$, which represents the density field of GMM $n(x)$, can be derived as
>
> $$n(x)=-\frac{\partial p(x)}{\partial x} = \sum_{m=1}^M \gamma\_m \Sigma_m^{-1}(x - \mu\_m),$$
>
> where $\gamma_m = \frac{\pi_m \mathcal{N}(x \mid \mu_m, \Sigma_m)}{\sum_{j=1}^M \pi_j \mathcal{N}(x \mid \mu_j, \Sigma_j)}$.
>
> ### **1. Our formulation of normal better reflects the property of local geometry compared to the negative gradient of superpositioned Gaussian field.**
>
> * **Discussion on $n(x)$**
>
> Based on the derivation above, we would like to point out that Gaussians cannot yield the universally correct normal based on the argument that "the normal should be the negative gradient of the density distribution" due to the **nature of a Gaussian function**. As can be seen, the key flaw of $n(x)$ in expressing geometry is that **$n(x)$ MUST BE directed outward from the gaussian center from our derivation**. (We have visualized $n(x)$ in 2D example via **`neg_grad_of_gaussian_viz.pdf`** in supplemenatry file or **[this link](https://drive.google.com/file/d/1-hjy5IIrD0saHZpVBBkF1uATqnL4MZep/view?usp=sharing)**) Thus, their composition should also be difficult to precisely reflect the local geometry. In fact, the original purpose of the Gaussian function in a splatting algorithm was a reconstruction kernel, or a monotonically decreasing weight function around a point density. Thus, it might be unnatural to interpret this simple weighting function as a complex density distribution function.
>
> **Let us further explain with a concrete example**. Consider the case where there are Gassians located on an underlying horizontal surface, and we are querying on the leftmost point for surface normal (**We have attached the illustration of this case via figure (a) of `comparison_to_field.pdf` in supplemenatry file or [this link](https://drive.google.com/file/d/1-Xpv-c6iEqzeM3GoKqn-feUDIkTIzrVi/view?usp=sharing) for better understanding**). In this case, if we calculate the normal using the gradient of neighboring GMM fields $n(x)$, normals calculated from $n(x)$ will be directed leftward. However, since our underlying surface is a horizontal plane, an ideal normal vector must be directed upward. Thus, we believe that using $n(x)$ or analytical gradient of GMM is rather imprecise compared to our expectation of precise geometry.
>
> Correspondingly, **`given that we are dealing with a Gaussian function as a primitive, there cannot exist a golden standard or a strictly correct way on defining normal that geometrically make sense`**. Thus, a strict mathematical derivation of the "error" (with respect to a "strictly correct way") of our rendering equation is difficult to exist at this point. In other words, we believe that the rendering equation proposed by RaDe-GS is rather a `formulation`, not approximation (because there cannot exist a "ground truth" to approximate in a **Gaussian field**).

---

> ### Author Response · Authors · 2024-11-22
> **Regarding the superposition of primitives (2/2)**
>
> * **How the rendering equation we leveraged is still a reasonable formulation.**
>
> In that context, we discuss how the rendering equation we adapted makes a reasonable formulation. Note that we cannot make a strict comparison, because there cannot exist an accurate $n(x)$ to compare with.
>
> **First**, we need to discuss whether the ray-Gaussian intersection is reasonable point to form a plane with. The intersection is the point that maximizes the Gaussian function among all points on the ray. Thus, this is the point where the ray is most likely to observe the Gaussian. Here, using $n(x)$ to calculate normal at this point will cause problems as mentioned above. Instead, our rendering equation locates another point where the density is most likely to exist in space, which is the center of Gaussian. Correspondingly, it is geometrically a reasonable formulation to form a plane with these points. (In fact, this formulation does cause view dependancy, as the reviewer well pointed out. We make a designated discussion section on this issue below.)
>
> **Second**, as our whole rasterization system uses affine approximation of projective trasnformation using the jacobian $J$, we apply the same to project the normal of the plane from ray space. Specifically, given a normal in ray space as $n'=(q \ \ 1)^{T}$ (plane formed by ray-Gaussian intersection and Gaussian center), $n$ is projected using the affine approximation as $n=Jn'$. **Thus, the error from our normal formulation is definitely created from this affine approximation via $J$, whose error gets larger as our query point for normal gets farther away from the Gaussian center**. We have visualized the approximated error via figure (b) of **`comparison_to_field.pdf`** in supplementary material. However, most works based on 3DGS and its affine approximation to projective transformation is not free from this error as well.
>
> * ## **Contribution of our work**
>
> First, we want to highlight that our paper is proposing a reasonable formulation of geomerically reasonable covariance, just as the recent rendering methods had to formulate a geometrically reasonable intersection plane. We believe that the formulation, instead of exact derivation, has been inevitable due to the nature of Gaussians, where its negative gradient field is not suitable for expressing geometry.
>
> In that context, **the key intuition our paper raises is that $n$ can also have error if $n'$ is imprecise even before projection**. We proposed methods to make $n'$ reflect its neighbors via aligning the covariance axis to the underlying surface. Otherwise, arbirarily rotated Gaussian will yield normal that is irrelevant to the underlying surface, as illustrated via figure (c) in **`comparison_to_field.pdf`** of supplementary material.
>
> ### **1.2. Then what would be a viable $p(x)$ where $-\frac{\partial p(x)}{\partial x}$ can geometrically make more sense?**
> One idea that we believe can work is the superposition of `sine` and `cosine`. According to the Fourier theorm, any function can be represented as an infinite sum of sine and cosine function. We believe that the superposition of these learnable periodic functions is a better approach to define $p(x)$, because:
>
> 1. Sum of sine and cosine functions can express more complicated density field.
> 2. Sine and cosine functions are easily differentiable, making it a computationally efficient choice for calculating normal field.
>
> We cannot agree more with the reviewer that strict definition and derivation of normal must be the ultimate goal to this field of study. However, we believe that it is possible `only when we can come up with p(x) whose n(x) makes sense`.

---

> ### Author Response · Authors · 2024-11-22
> **Regarding the anisotropic nature of Gaussians vs. kNN | Regarding to the statement on our third contribution**
>
> >
> > ## Regarding the anisotropic nature of Gaussians and kNN
> >
>
> We agree with the reviewer that considering only the means of neighboring Gaussians using kNN may not always be stable as kNN may exclude relevant Gaussians or include irrelevant Gaussians as neighbor, primarily because it does not consider the anisotropic nature of neighboring Gaussians.
>
> However, we would like to make two points that kNN can reasonably work in our method.
>
> ### 1.  **What we have proposed can cover possible instability that kNN brings.**
>
> Two of our proposed methods can behave to make our method less unstable when using kNN:
>
> a. **residual rotation**
>
> In cases when eigenmatrix become inaccurate due to the inclusion of irrelevant Gaussian as the neighbor, we introduced a residual rotation $\Delta R$ so that our final covariance orientation can be jointly expressed with it as $V' = \Delta R V$. Thus, $\Delta R$ will compensate for imprecise $V$ during optimization.
>
> In other words, it is reasonable to infer that the larger $\Delta R$ is, the more imprecise $V$ tend to be, which can imply that kNN is performing badly. However, we showed from our appendix that most $\Delta R$ s are very close to 0, meaning that kNN brought reasonable set of neighbors for most Gaussians.
>
>
> b. **local planarity for eigenvector regularization**.
>
> Even if we adjust the eigenmatrix using $\Delta R$, its realiability still cannot be assured if included with unnecessary neighbors. Thus, we quantified  the reliability using the local planarity $\gamma$ we proposed in Eq.(10). When constructing eigenvector similarity loss $\mathcal{L}\_{\text{eigval}}$, we compared adjacent neighboring eigenvectors proportional to this local planarity. In that way, neighbors with higher uncertainty will receive less optimization feedback.
>
> To summarize, we agree with thre reivewer's concern that applying kNN on Gaussians may bring instability. However, we believe that our residual rotation and local planarity can soothe down these drawbacks.
>
>
> ### 2. **Despite the possible drawbacks mentioned by the reviewer, recent works on 3DGS demonstrate that kNN can be employable to 3D Gaussians**
>
> We totally understand the drawback the reviewer pointed out on kNN. However, we would still like to note some of the recent works that have utilized kNN of Gaussian means and solved their problem definitions. For instance, **Structure-Aware Gaussian Splatting [F]** trains a graph neural network of Gaussians, whose edges are constructed by kNN of Gaussian means. Also, **CoR-GS [G]** utilizes kNN to find the matching correspondances between two sets of Gaussians. These recent works show that kNN does get along with the Gaussians if used properly.
>
>
> >
> > ## **Regarding to the statement on our third contribution**
> >
>
> ### **On stating that ours *"maintain faster training time compared to implicit neural representation-based method"***
>
> We intended to state that our work is faster than the aforementioned **implicit** neural representation-based methods such as NeRF, NeuS, VolSDF and Neuralangelo that take >12h for training while ours take $91.2$ min.
>
> ### **On stating that *"Our method achieves state-of-the-art performance in surface reconstruction tasks among 3DGS based methods"***
>
> While we acknowledge that ours achieves slower training time than other 3DGS-based methods, ours outperformed Chamfer Distance (CD) compared to these recent 3DGS frameworks. We apologize for the confusion by using a vague and comprehensive term such as *performance*, **which can misguide the readers that ours also achieves the fastest training time.** We will rephrase the third contribution by specifying that ours yield comptetitive surface reconstruction quality among 3DGS-based methods.
>
> We thank the reviewer for recognizing PGSR. We have toned down the third contribution accordingly.
>
> Meanwhile, we would like to point out that PGSR proposes depth and normal rasterization method given a Gaussian followed their regularization in screenspace. However, we believe that our paper is proposing an orthogonal question, which highlights the importance of **aligning of the Gaussian orientations to surfaces in 3D** that PGSR is not tackling. Thus, we believe that our research paper can make a different yet novel impact compared what PGSR makes in 3DGS community.
>
> ***Reference***
>
> **[F]** Ververas et al. "SAGS: Structure-Aware 3D Gaussian Splatting.", ECCV 2024
>
> **[G]** Zhang et al. "CoR-GS: sparse-view 3D Gaussian splatting via co-regularization." ECCV 2024

---

> ### Comment · Reviewer_2Feu · 2024-11-26
> **The algorithm should be stronger**
>
> I appreciate the efforts. However, my concerns remain unresolved. The explanation provided did not address the flaws, and the additional results do not consistently support the effectiveness of the method. I am more convinced that the method is not ready for publication. Here are my thoughts.
>
> 1. The method shows no improvement on Tanks and Temple, a common benchmark dataset. One of the key challenges is the selection of K in KNN, which impacts the performance. When applied to larger scenes, the method that is optimized for object-level datasets proves to be ineffective. This is likely due to the Gaussian primitive distribution in open scenes being less uniform. Furthermore, the authors did not provide information on the training time for TnT, which leads me to believe that the method’s costs would be greater in such datasets.
>
> 2. The method still heavily relies on other regularizations, which makes me feel like the contribution is more marginal. I was somewhat misled by the introduction because I don't see any dependencies. I feel like bringing locality into 3D Gaussian should at least demonstrate better results but it shows no. Appendix A.1 did not fix this question; it only explained that the L_{eigvec} is better than L_n, but not dependencies. If it is better, why does it still rely on it? If the L_{eigvec} promotes local planarity, then the depth normal should already align with the render normal. However, excluding {L_n} shows almost no improvement. If your method is incremental to existing regularization techniques, should you analyze more on how they are insufficient and how you address them?
>
> 3. The authors claim the contribution is a new formulation, not a theory,  and there cannot exist a golden standard or a strictly correct way of defining normal that geometrically makes sense. This makes me feel like the current method is an empirical study without a clear basis. But a scientific paper should clearly state that the formulation is more accurate.
>
> In summary, the current empirical study does not meet my expectations for publication at ICLR. The cost of the proposed method (the training speed) outweighs its benefits. Additionally, the technique lacks a clear explanation, which hinders further exploration by the community. Furthermore, the results do not show impressive performance across various datasets. To improve the paper,
> 1. Try to demonstrate the proposed method that is more accurate in a mathematical aspect.
>
> 2. Make the algorithm stronger and more efficient. Showing good results on a small dataset is not enough. There is still a challenge in applying this method to other datasets.
>
> 3. (Minor) Better articulating the paper’s aims and contributions. If the paper focuses on a regularization technique that builds upon existing approaches, it is essential to analyze how those existing methods fail and explain how the proposed method addresses these shortcomings. For instance, if the technique is a plug-and-play regularization method, it should apply to all existing techniques (e.g., 2DGS, GOF, RadeGS) with clear evidence of the benefits. Building upon the best baseline delivers increased performance, but no insight. Also, try to rephrase the contribution. I do not see any point in claiming the method is faster than implicit methods because you have built upon GS-based methods while making them significantly slow. Implicit methods should not be considered the primary competition.

---

> > ### Author Response · Authors · 2024-11-27
> > **[2/3] Author's response**
> >
> > >
> > > ## 2. **On negligible improvements on unbounded scene such as TnT**
> > >
> >
> > We agree with the cause conjectured by the reviewer on non-uniformity of unbounded scene. Note that we have also made a detailed discussion on another possible cause of this limitation in unbounded scenes at `2. Failure case & limitations` section in the `General Response`.
> >
> > Nonetheless, We believe a research paper is worth publication if it proposed a novel method and made reasonable proof-of-concept. It is difficult for novel ideas to be mature and work in all scenarios from the first research effort.
> >
> > *For instance, initial algorithms of NeRF did not successfully handle unbounded scenes. It was the following research efforts such as NeRF++***[H]** *or mipNeRF-360 that has made specific investigations and successfully addressed unbounded scenes for NeRF.*
> >
> > In that persepctive, we think that our work made a reasonable proof-of-concept that ensuring 3-dimensional local alignment of Gaussians IS effective, which was provable under an assumption that the scene is not unbounded. We hope that future works can address the unbounded scene problem while keeping the advantages of our parameterization. We believe this is possible because our work **did not deteriorate but preserved the quality on unbounded scenes**.

---

> > ### Comment · Reviewer_2Feu · 2024-11-27
> > **Immature and low impact are reasons for no acceptance**
> >
> > I appreciate the authors' detailed responses. I was also pleased to see that the author admitted **the current method has less-established math, offers marginal improvement, consumes more training time, and cannot be applied to larger scenes.**
> >
> > I understand the author's complaint about this subject matter. Yes, scientific review is somewhat dependent on the reviewer's knowledge level. You might consider it a random review if it doesn't rate your paper highly. However, from my expertise, I have my key aspects to evaluate a paper.
> >
> > Now let's focus on "the marginal improvement (CD~0.02) outweighs its disadvantages (5x training cost, failure or no improvement in open scenes).
> >
> > -----------
> >
> > **1. Regarding cost versus benefit, unbounded scene and training costs are quite important for GS-based surface reconstruction methods.**
> >
> > Here are some points with which I respectfully disagree.
> > >  we believe that our proposed method may bring relatively smaller but non-negligible improvements compared to the previously proposed method.
> >
> > If users are considering object-level reconstruction, why not choose SDF-based methods like [1,2,3,4] with Instant-NGP or PermutoSDF as a backbone? These options are faster and more accurate than your method.
> >
> > Additionally, the smoothness of the surface is also negligible, as numerous surface reconstruction methods utilize monocular priors, often with minimal cost [4,5]. Previous works demonstrate that these techniques are significant. I also regret that you didn’t cite and compare them.
> >
> > >  It is difficult for novel ideas to be mature and work in all scenarios from the first research effort. For instance, initial algorithms of NeRF did not successfully handle unbounded scenes. It was the following research efforts such as NeRF++ or mipNeRF-360 that has made specific investigations and successfully addressed unbounded scenes for NeRF.
> >
> > I respectfully disagree. Both MipNeRF-360 and NeRF++ enable the baseline NeRF model in ways it cannot achieve, while your algorithm disables the capabilities of the RaDe-GS that it is capable of accomplishing.
> >
> > When improving an algorithm in one area, it's essential to consider the associated costs; otherwise, it's better to explore alternative solutions. I believe the techniques introduced in references [4] and [5] could also address this issue. It doesn’t make sense to claim that future research will be conducted to enhance your approach. If you know how to make improvements, why not implement them and resubmit?
> >
> > -------
> > **2. On the claim. You have built upon GS-based methods while making them significantly slow. Implicit methods should not be considered the primary competition, considering you do not discuss most implicit-based works.**
> >
> > > We have made this claim especially for readers with few background knowledge that training time for rasterization-based method, including ours, is faster than ray-tracing based implicit methods.
> >
> > I strongly disagree with this response. A scientific paper should clearly structure its claims and contributions rather than mislead readers. I have noticed that `Review XitP` has copied the misleading contribution in `Strength 2` and given it a higher rating. Readers with little background knowledge can be more easily misled rather than gaining a true understanding.
> >
> > > Review XitP S2: Quantitative and qualitative results on multiple datasets demonstrate the effectiveness of the proposed method -- it achieves new state of the art in surface reconstruction among 3DGS-based methods while maintaining faster training time compared to implicit neural representation-based methods.
> >
> > I believe that while paper always has its place, this one is definitely not suitable for ICLR until those points are addressed.
> >
> > [1] Geo-Neus: Geometry-consistent neural implicit surfaces learning for multi-view reconstruction
> >
> > [2] NeuS2: Fast Learning of Neural Implicit Surfaces for Multi-view Reconstruction.
> >
> > [3] Voxurf: Voxel-based Efficient and Accurate Neural Surface Reconstruction
> >
> > [4] PermutoSDF: Fast Multi-View Reconstruction with Implicit Surfaces using Permutohedral Lattices
> >
> > [4] Monosdf: Exploring monocular geometric cues for neural implicit surface reconstruction.
> >
> > [5] Reducing Diffusion Variance for Stable and Sharp Normal.

---

> > > ### Author Response · Authors · 2024-11-28
> > > **[1/3] Regarding benefit vs. cost**
> > >
> > > **We appreciate the reviewer's detailed feedback.**
> > >
> > > As far as we understood, the keys of the reviewer's query is
> > >
> > > 1. Cost vs. benefit of our method, and
> > >
> > > 2. The contribution statement
> > >
> > > We will try our best to fix any misunderstandings and share our thoughts.
> > >
> > > ---
> > >
> > > * **(0-1) Remarks on benefits**
> > >
> > > **`Please also acknowledge the visual improvements that CD gain of 0.02 brings when balancing the cost and benefit.`**
> > >
> > > The visual improvements presented in the result figures is a reminder that the *seemingly small number of 0.02 is in fact significant*, as discussed in the **General Response**.
> > >
> > >
> > > * **(1) Reviewer's Q: Why not choose SDF-based methods like [1,2,3,4] with Instant-NGP or PermutoSDF as a backbone? These options are faster and more accurate than your method.**
> > >
> > > We designed the objective of our work to ***investigate whether achieving 3D continuity in Gaussian Splatting will bring improvements in its surface reconstruction quality.*** That is why we made a specific investigation on 3DGS-based methods.
> > >
> > > Indeed, integation of SDF-based methods like [1,2,3,4] with faster training methods such as Instant-NGP or PermutoSDF as a backbone is another great research direction to achieve both accuracy and training speed.
> > >
> > > * **(2) Reviewer's Q: Additionally, the smoothness of the surface is also negligible.**
> > >
> > > We respectfuly believe that we have demonstrated clear improvements in broken geometry caused by non-smooth alignment of local Gaussians (i.e., forehead of skull in Fig. 1, stomach of teddy bear in Fig. 4, floor in Fig. 5 and body of mic in Fig. 6) compared to competitive 3DGS baselines (all of which include $\mathcal{L}\_{d}$ and $\mathcal{L}\_{n}$).
> > >
> > > * **(3) Reviewer's Q: Numerous surface reconstruction methods utilize monocular priors, often with minimal cost [5, 6]. Previous works demonstrate that these techniques are significant.**
> > >
> > > Note that we did cite monoSDF [5] in related works, but did not compare with our work because monoSDF leverages external knowledge from model trained with large training data, making the comparisons unfair. That is why we have made the list of comparing works that leverage no prior knowledge.
> > >
> > > ---
> > >
> > > * **`(4) Our thoughts on balancing between benefits versus cost`**
> > >
> > > We understand that the reviewer make heavier emphasis on our limitation in training time and limited scene size over the benefits our method bring. We respect the reviewer for having firm and clear standard, and acklowledge that the reviewer is simply doing his/her job.
> > >
> > > However, we also want to share our findings to audience with different standards: people who are interested in **methods for quality improvement in 3DGS for bounded scenes over its training time**. Thus, we think it is beneficial to 3DGS community to gain the visibility of our work.
> > >
> > >
> > >
> > > >
> > > > ***Reference***
> > > >
> > >
> > > [1] Geo-Neus: Geometry-consistent neural implicit surfaces learning for multi-view reconstruction
> > >
> > > [2] NeuS2: Fast Learning of Neural Implicit Surfaces for Multi-view Reconstruction.
> > >
> > > [3] Voxurf: Voxel-based Efficient and Accurate Neural Surface Reconstruction
> > >
> > > [4] PermutoSDF: Fast Multi-View Reconstruction with Implicit Surfaces using Permutohedral Lattices
> > >
> > > [5] Monosdf: Exploring monocular geometric cues for neural implicit surface reconstruction.

---

> > > ### Author Response · Authors · 2024-11-28
> > > **[3/3] Fixing possible misunderstandings.**
> > >
> > > We would respectfully clarify any possible misunderstandings on our claim.
> > >
> > >
> > > ## **1. We resepectfully did not intend to claim that our method has less-established math.**
> > >
> > >
> > > First, we would like to revisit some possible misunderstanding from the reviewer.
> > >
> > > > `I was also pleased to see that the author admitted the current method has less-established math`
> > >
> > > We apologize for the confusion. We may kindly refer to the constructive discussions we had to check any source of misunderstandings.
> > >
> > > * **Summary of our conversation**
> > >
> > > 1. The reviewer kindly requested a strict mathematical comparison of normal rendered from our work with the reviewer's proposal of an ideal definition of normal, which is the negative gradient field of superposition of Gaussian functions.
> > >
> > > 2. So, we mathematically derived the negative gradient field of Gaussians to do so. During the process, we learned otherwise that the gradient field of superpositioned Gaussians is in fact not geometrically plausible, making it unsuitable for reference of accuracy analysis.
> > >
> > > 3. We showed that it is due to the monotonically decreasing nature of Gaussian function that greately limits Gaussian from complex expression of density field. We have made deailed discussion on this reason.
> > >
> > > 4. Thus, we concluded that it is not viable to compare our rendering equation with the negative gradient field of Gaussian function.
> > >
> > > **We may kindly note that the our conclusion in 4. does not conlcude that the rendering equation has "less-established math."**
> > >
> > > **As the best alternative, we highlighted the mathematcal theory of our rendering equation by explaining**
> > >
> > > > *"The rendering equation forms a plane using points with the maximum likelihood of observation (conveniently called as ray-Gaussian intersection) and the maximum likelihood of existence (Gaussian center). Consequently, this plane (thus its normal) is the interpretation of a Gaussian geometry with the most likelihood."*
> > >
> > > We believed that this could better explain the grounds of our rendering equation as a geometrically and probabilstically reasonable framework.
> > >
> > > We apologize if our answers made confusions and could not satisfy the reviewer's need.
> > >
> > >
> > > ### * **Further emphasis on the core of our work** (What we believe is the most important)
> > >
> > >
> > > The core mathematical background of our method lies on achieving 3D continuity of Gaussians using eigenvectors and eigenvalues as means of parameterization and regularization. ***We kindly ask the reviewer to also take consideration on this aspect as well.***
> > >
> > > --
> > >
> > >
> > > ## **2. We respectfully did not intend to claim that our method offers marginal improvements.**
> > >
> > >
> > > We would also like to fix another misunderstanding
> > >
> > > > `I was also pleased to see that the author admitted the current method offers marginal improvements`
> > >
> > > We apologize for the confusion. We have tried to consistently claim **otherwise** that the improvements we have made is not marginal. Summary on this claim can be found at "(1) On "seemingly" negligible improvement in quantitative measures" of the general response. *(Balancing this reasonable improvement in surface reconstruction quality with the cost of more training time is discussed on the previous response.)*
> > >
> > > We conjecture that **we falsely misled the reviewer** by us saying:
> > >
> > > >We believe that our proposed method may bring relatively smaller but non-negligible improvements compared to the previously proposed method.
> > >
> > > According to the context of discussion that we referred to the "previously proposed method", we were referring to the previous regularization methods in 3DGS-based baselines, $\mathcal{L}\_{d}$ and $\mathcal{L}\_{n}$. Our message was : despite that improvement contributed by our regularization is not as much as the previously proposed regularization, our regularization still made a reasonably positive impact. **`Please note that we got the best results when ours and the existing regularization methods are both applied.`**
> > >
> > > Thus, we kindly claim that the relative comparison of our work to the previous work ($\mathcal{L}\_{d}$ and $\mathcal{L}\_{n}$) does not make our work to be marginal.
> > >
> > > **We thank the reviewer for the dedication.**

---

> ### Author Response · Authors · 2024-11-27
> **[1/3] Author's response**
>
> **`First of all, we deeply appreciate the dedication, time and effort the reviewer is making.`**
>
> We would like to answer the questions below!
>
> >
> > ## **1-1. On geometrical "accuracy" of the adopted rendering equation (RaDe-GS) in mathematical aspect.**
> >
>
> *Given a Gaussian, a ray, and the normal rendering equation, **with what "ground-truth" normal expression** shall we mathematically compare with to demonstrate rendering equation's accuracy? The reviewer initially suggested the negative gradient of Gaussians, which we tried, analyzed and illustrated with examples that it is not a suitable representation for ground-truth surface normal. We instead proposed an alternative research direction on defining a precise ground-truth density field model using the superposition of sine and cosine functions based on the Fourier theorem. However, this is a completely new idea and is worthy of a new research topic.*
>
> That is why we would like to highlight and stay consistent to the point: Given that we have to deal with a `Gaussian function`, a "ground-truth" way of defining normal cannot exist at the moment. In fact, this is why it is hard to demonstrate mathematical accuracy of ***ANY*** rendering equations given Gaussian primitives.
>
> Thus, the research question should instead be **defining a rendering equation that makes a geometrically reasonable interpretation of Gaussians**.
>
> In that perspective, we believe that RaDe-GS has proposed a rendering equation with reasonable basis.
>
> *RaDe-GS forms a plane using points with the **maximum likelihood of observation** (conveniently called as ray-Gaussian intersection) and the **maximum likelihood of existence** (Gaussian center).* Consequently, this plane (thus its normal) is the interpretation of a Gaussian geometry with the most likelihood.
>
> ---
>
> * **Reviewer's great suggestion on experimenting our method on 2DGS and GoF to show that our method brings improvements regardless of the rendering equation.**
>
> We need to show that applying our method should bring improvements on other rendering equations in order to make our empirical evidence stronger. We thank the reviewer for this valuable insight.
>
> We have experimented our methods with rendering equations proposed by **2DGS** and **GOF**. For 2DGS, CD improved from $0.75$ to $0.72$, while for GOF, CD improved from $0.72$ to $0.70$. We have updated the appendix of our manuscripts with CD for each scene in DTU dataset for 2DGS and GOF.
>
> We believe that these experimental result along with the experiments in our manuscript demonstrates the effectiveness of our parameterization and regularization method.
>
> >
> > ## 1-2. **The basis of our research is clear: *Eigenvectors to align Gaussians for smooth continuity in 3D***
> >
>
> Here, we would like to refresh our discussion to focus on our key contribution. We would like to highlight that the core of our proposed method is the **alignment of Gaussians for smooth continuity in 3D, which is achieved via parameterization of Gaussian Covariance using eigenvectors**.
>
> Of course, there IS a mathematical ground on determining the orientatation of Gaussian covariance using eigenvectors of adjacent Gaussian means, where the third eigenvector can be mathematically regarded as the representation of this local orientation, because it is the direction of the smallest variance of distribution. We have explained the mathematical background on this design in the paper, which we believe that the reviewer is well aware of in this aspect.

---

> ### Author Response · Authors · 2024-11-27
> **[3/3] Author's response**
>
> >
> > ## **3. Further articulation of our regularization methods and their contribution**
> >
>
>
> * **Why we need *both* our regularization and the existing regularization, $\mathcal{L}\_{n}$ and $\mathcal{L}\_{d}$ ?**
> ---
>
> *We will make sure to add this discussion in the appendix of our manuscript. We thank the reviewer for the valuable insight.*
>
> In the appendix, we tried to distinguish the different working mechanism between the existing regularization and our regularization method. However, it was not intended to state that one is better than another, or else one must be replacable by another as the reviewer correctly pointed out. In fact, we need to inform the readers on why we need both, as the reviewer correctly stated.
>
> The key reason for requiring both is that the **existing regularization method is conducted in 2D screen-space, while ours is conducted in 3D world space**. This is a big difference because the geometry in screen-space is the volume-rendered depth/normal map, while our method is a direct regularization of raw 3D primitives "before" rendering.
>
> Thus, the underlying dynamics of the two losses must be different, making the two losses complementary. **For instance, the existing regularization affects Gaussians distributed along a ray, while our regularzation affects K nearest Gaussians per Gaussian indenpendent from a ray.**
>
> We additionally showed from the previous reply titled `Further empirical studies` that ours and the existing regularzations are indeed complementary and yield the best empirical results when used together.
>
>
>
>
> * **Answer to: *"The method still heavily relies on other regularizations, which makes me feel like the contribution is more marginal"***
> ---
>
> Based on the answer above and the experimental result from the previous reply titled `Further empirical studies`, we believe that our proposed method may bring *relatively smaller but **non-negligible** improvements comp    ared to the previously proposed method*.
>
> Above all, all 3DGS-based methods except SuGaR uses the existing regularization $\mathcal{L}\_{n}$ and $\mathcal{L}\_{d}$, which **by themselves still yield the broken geometries displayed in the reported figures.** Also, even though the CD improvments of `0.02` ~ `0.03` may seem negligible, we have made a discussion in `(1) On seemingly negligible improvement in quantitative measures.` of General Response to show that the improvements driven **purely** by our method is **not** negligible.
>
>
> * **Answer to: *"I feel like bringing locality into 3D Gaussian should at least demonstrate better results but it shows no."***
> ---
> *Surface quality-wise*: It is difficult to agree that our reported results are not better compared to the results of recent 3DGS baselines. We believe that we have demonstrated clear improvements in broken geometry caused by non-smooth alignment of local Gaussians (i.e., forehead of skull in **Fig. 1**, stomach of teddy bear in **Fig. 4**, floor in **Fig. 5** and body of mic in **Fig. 6**) compared to competitive 3DGS baselines.
>
> * **Answer to: *"The cost of the training speed outweights its benefits."***
> ---
> We acknowledge that ours take relatively longer training time than 3DGS baselines. *However, we believe that it is rather a subjective matter whether surface reconstruction quality outweights slower training time, or vice versa.*
>
> In other words, it is up to the scenarios of many users, where it is reasonable to believe that many of them would choose ours over RaDe-GS by sacrificing 75 minutes of additional training time for better reconstruction quality, while implicit methods for >12h is way too long.
>
> * **Answer to: *I do not see any point in claiming the method is faster than implicit methods because you have built upon GS-based methods while making them significantly slow.***
> ---
> We have made this claim especially for readers with few background knowledge that training time for rasterization-based method, including ours, is faster than ray-tracing based implicit methods.
>
> *For instance, our method has no use if our method took >12h training time while keeping the same surface reconstruction results as it is now.*
>
> Also, we want to make sure that for the inference time, **FPS of our method is identical to 3DGS** (~ 130 FPS in RTX3090). That is because $V'$ is calculated only one time and replace $R$ right before the storage of Gaussian .ply file.
>
> >
> > ***Reference***
> >
>
> **[H]** Zhang, Kai, et al. "NeRF++: Analyzing and improving neural radiance fields."

---

> ### Author Response · Authors · 2024-11-28
> **[2/3] Regarding the contribution statement**
>
> ## **Reviewer's Q: Implicit methods should not be considered the primary competition**
>
>
> We agree with the reviewer that implicit-based methods are not being considered as the primary competition in our work. Especially, the purpose of our work is to investigate whether achieving 3D continuity in Gaussian Splatting will bring improvements in its surface reconstruction quality. Thus, it is more relevant to make more focus on 3DGS-based method.
>
>
> ## **Reviewer's Q: Isn't the contribution statement misleading?**
>
>
> We would like to summarize the worries of the reviewer, followed by our intention on the statement for clear and organized discussion.
>
> First, here is our latest contribution statement (edited version in the current manuscript after the reviewer's comment):
>
> > Our method achieves competitive quantitative and qualitative results in surface reconstruction among 3DGS-based methods, while maintaining faster training time compared to implicit neural representation-based methods.
>
> --
>
> ### * **What the reviewer is concerned to be misleading**
>
> From the best of our understanding, the reviewer is concerned of the misleadment based on the following
>
> 1. Ours yield the slowest traing time compared to all 3DGS-based method.
>
> 2. Stating that ours is faster than implicit-based method may misleadingly conceal the fact in 1.
>
> 3. In addition, implicit methods are not primary works of competition. Thus, they may not be referenced.
>
> ***Please correct us if we misunderstood the reviewer's concern***
>
> --
>
> ### * **Our intention on claiming the contribution statement**
>
> From the contribution statement, we wanted to make sure that
>
> 1. Ours bring competitive results in surface reconstruction among 3DGS-based methods.
>
> 2. The competitive result comes with the training time above the implicit method's training time.
>
> 3. Implicit methods should be mentioned as a secondary, referential lower bound. For example, ours winning surface reconstruction quality among 3DGS-based methods while taking similar training time with implicit-based methods (>12h) is **meaningless.**
>
> *(If our method do take >12h for training, our method may not even be worthy of discussion for the benefit vs. cost. **Because in this case, ours loose competitiveness to Neuralangelo in both result and speed**)*
>
> --
>
> ## **Rephrasing the contribution statement**
>
> Even though we believe that our original statement is not false, we understand that the statement may mislead the readers that our work make a thorough study with implicit methods as a primary comparison. Considering that the readers can still have information from Table. 1 that ours is still faster than many implicit methods, we would like to rephrase the contribution as below;
>
> >  Our method achieves competitive quantitative and qualitative results in surface reconstruction among 3DGS-based methods.
>
> ***We would kindly ask the reviewer whether this statement clears the possible misunderstanding.***

---

> ### Comment · Reviewer_2Feu · 2024-11-28
> **Response**
>
> Over the past year, significant efforts have been made in GS-based surface reconstruction. SuGaR [1] first estimates the Signed Distance Function (SDF) but has proven to be less efficient and accurate due to the imprecise k-nearest neighbors (KNN) used for Gaussians. Subsequent researches have improved this line of research, either through better representations or regularizations: some studies have focused on the normals of Gaussians along their shortest axis [1, 3, 5] or directly utilized 2D Gaussians [2], as the direction corresponds to where the volume densities decrease most rapidly, which aligns well with NeRF-based methods; other research has shown that screen space techniques, including depth-normal losses [2], and multiview-consistency losses [3] are more effective and efficient.
>
> However, this work seems to step backward the progression of GS-based surface reconstruction, raising questions about its value. It also relies on a non-peer-reviewed paper [9] for surface normal estimation without adequate grounding, making me wonder whether this work prioritizes achieving state-of-the-art (SOTA) results over correctness. The authors claim that their GS-based method is SOTA (which is false) and faster than implicit methods in terms of training speed. Yet, the fastest surface reconstruction technique to date remains grid-based SDF [6,7,8]. Such claims undermine our community rather than provide readers with valuable knowledge, except attracting reviewers for higher ratings.
>
> From this perspective, I believe that using certain hacks to achieve marginal results for a CV task may not align well with ICLR's scope. **I will not respond anymore and suggest leaving the decision to AC or other reviewers.**
>
> --------
> [1] SuGaR: Surface-Aligned Gaussian Splatting for Efficient 3D Mesh Reconstruction and High-Quality Mesh Rendering. CVPR'24
>
> [2] 2D Gaussian Splatting for Geometrically Accurate Radiance Fields. SIGGRAPH'24
>
> [3] PGSR: Planar-based Gaussian Splatting for Efficient and High-Fidelity Surface Reconstruction. TVCG'24
>
> [4] GaussianShader: 3D Gaussian Splatting with Shading Functions for Reflective Surfaces. CVPR'24
>
> [5] High-quality Surface Reconstruction using Gaussian Surfels. SIGGRAPH'24
>
> [6] NeuS2: Fast Learning of Neural Implicit Surfaces for Multi-view Reconstruction.
>
> [7] Voxurf: Voxel-based Efficient and Accurate Neural Surface Reconstruction
>
> [8] PermutoSDF: Fast Multi-View Reconstruction with Implicit Surfaces using Permutohedral Lattices
>
> [9] RaDe-GS: Rasterizing Depth in Gaussian Splatting. arXiv:2406.01467

---

> ### Author Response · Authors · 2024-11-29
> **Response to the final comment from the reviewer**
>
> First of all, we would like to thank the reviewer 2Feu for the dedication he/she made.
>
> ## **The reviewer made some constructive comments that led us to improve our manuscript, such as**
>
> 1. **Experiments to show that our method brings improvment to other rendering equations such as 2DGS [1] and GOF [2].**
>
> *(Results reflected to manuscripts to Appendix A.7 in Nov. 27)*
>
> 2. **Rephrasing the contribution of our work to achieve *"competitive quantitative and qualitative results in surface reconstruction"* instead of *"SOTA"* in surface reconstruction.**
>
> *(Manuscript correspondingly edited on p.3 l.115 in Nov. 25)*
>
> --
>
> ## **We make our comment to the reviewer's final response.**
>
> * *`(1) "making me wonder whether this work prioritizes achieving state-of-the-art (SOTA) results over correctness."`*
>
> We would like to emphasize the mathematical background of the rendering equation we adopted.
>
> > *"The rendering equation forms a plane using points with the maximum likelihood of observation (conveniently called as ray-Gaussian intersection) and the maximum likelihood of existence (Gaussian center). Consequently, this plane (thus its normal) is the interpretation of a Gaussian geometry with the most likelihood."*
>
> Most importantly, the **main focus** of our design is rather about **parametrizing covariance with eigenvectors of distribution of neighboring Gaussians' mean**. We aimed for both performance and the logical background of our work, which the rest of the reviewers approved the sanity of our new approach. We respectfully do not believe that such grounds of our work are incorrect.
>
> * *`(2) "The authors claim that their GS-based method is SOTA (which is false)"`*
>
> We would like to respectfully remind our actual words of statements. We stated in our updated manuscript as:
>
> > Our method achieves competitive quantitative and qualitative results in surface reconstruction among 3DGS-based methods,
>
> *(Updated on Nov. 22)*
>
> * *`(3) "The authors claim that their GS-based method (is) faster than implicit methods in terms of training speed. Yet, the fastest surface reconstruction technique to date remains grid-based SDF [3,4,5]"`*
>
> The implicit methods that we were referring to were ones (i.e., Neuralangelo: 0.61 CD) that greately outperform our method (0.67 CD), rather than ones that yield slightly worse or comparable CD (NeuS2 [3]: 0.70, Voxurf [4]: 0.72, PermutoSDF [5]: 0.68, *all values are referenced from the papers*) compared to ours.
>
> However, the reviewer's concern had its point. **Thus, we make a kind reminder that we have promised from the previous response to rephrase our contribution statement by removing the comparison of training time with implicit methods to ease any misguidance**. We thank the reviewer for suggesting the valuable insight.
>
> Most of all, both the reviewer and we agreed that **the primary comparison must be 3DGS-based methods, since our work is an investigation on local continuity of Gaussians in 3D.**
>
> * *`(4) "It also relies on a non-peer-reviewed paper [6] for surface normal estimation without adequate grounding."`*
>
> Despite the fact, RaDe-GS[6] is a well established [codebase](https://github.com/BaowenZ/RaDe-GS) with widespread use. We have also explained its mathematical background:
>
> > *"The rendering equation forms a plane using points with the maximum likelihood of observation (conveniently called as ray-Gaussian intersection) and the maximum likelihood of existence (Gaussian center). Consequently, this plane (thus its normal) is the interpretation of a Gaussian geometry with the most likelihood."*
>
> We believe that the prevalence of codebase and its mathematically reasonable background cannot undermined.
>
> ---
>
> ## **Here is the final summary of our work**
>
> 1. Our work tackles the problem specific to 3DGS, where the covariance of Gaussians can be oriented arbitrarily, often overlooking the distribution of neighboring Gaussians.
>
> 2. Our work proposes a method for achieving 3D continuity by orienting Gaussians via eigenvectors/values of neighboring Gaussian centers.
>
> 3. Our work clearly demonstrate its effect on surface reconstruction results both quantitatively and qualitatively.
>
> We again thank the reviewer for the valuable responses.
>
> Best,
>
> The authors.
>
>
> ***Reference***
>
>
> [1] 2D Gaussian Splatting for Geometrically Accurate Radiance Fields. SIGGRAPH'24
>
> [2] Gaussian Opacity Fields: Efficient Adaptive Surface Reconstruction in Unbounded Scenes. SIGGRAPH ASIA'24
>
> [3] NeuS2: Fast Learning of Neural Implicit Surfaces for Multi-view Reconstruction.
>
> [4] Voxurf: Voxel-based Efficient and Accurate Neural Surface Reconstruction
>
> [5] PermutoSDF: Fast Multi-View Reconstruction with Implicit Surfaces using Permutohedral Lattices
>
> [6] RaDe-GS: Rasterizing Depth in Gaussian Splatting.

---

### Official Review · Reviewer_amX1 · 2024-10-31

**Soundness:** 3
**Presentation:** 3
**Contribution:** 3
**Rating:** 6
**Confidence:** 4

**Summary:**

This paper focuses on constructing high-quality geometric surfaces in 3D Gaussian Splatting (3DGS) representation, with depth and normal learning being key aspects. Unlike existing methods that learn normals for each Gaussian individually, this paper proposes deriving normals using the distribution of neighboring Gaussians. Validated across multiple datasets, this approach enables the paper to achieve smoother local surfaces.

**Strengths:**

● This paper is a comprehensive work that starts by addressing issues arising from existing methods that optimize each Gaussian individually. It proposes a normal expression based on local regions and designs several regularization methods based on this normal computation.

● The starting point of the paper is indeed one of the key differences between the existing 3DGS framework and traditional surface reconstruction, making it quite insightful. Traditional surface reconstruction places greater emphasis on the distribution of point clouds in the neighborhood, while existing 3DGS frameworks rely more on 2D rendering loss, often overlooking 3D continuity.

● The experimental validation in this paper is thorough, including both quantitative and qualitative comparisons with existing methods on mainstream datasets, as well as ablation studies of the proposed regularization losses.

● The paper’s presentation is very clear and easy to read.

**Weaknesses:**

● With the use of the proposed covariance design and additional regularization losses, the improvement over the existing optimal solution is not significant; on the DTU dataset, the Chamfer Distance (CD) only improves by 0.02 compared to RaDe-GS, while the training time is 75 minutes longer (more than five times that of RaDe-GS).

**Questions:**

● If the PCA-based regularization on local regions proposed in the paper is directly introduced into the existing 3DGS framework, can it also achieve similar smoothing effects as described in the paper?

● As far as I know, in traditional point cloud-to-surface reconstruction, the size of the local region has a significant impact on the final reconstruction results. Does the value of k in the kNN used in the paper also have a substantial influence? Should it be individually adjusted for scenes with varying levels of surface complexity?

---

> ### Author Response · Authors · 2024-11-19
>
> **We thank the reviewer for acknowledging our work. We especially appreciate the question on relating K to varying size and complexity of local region.**
>
> >
> >## $K$ for varying size of local region
> >
>
> We thank the reviewer for pointing out this important question. To answer to the question, we would like to first define an intuitive way of estimating the *size* of a local region, what hyperparameter determines the size and why, and finally how $K$ relates to the local size, thus the hyperparameter.
>
> First, the expected size of a local region can be approximated by the expected distance between two adjacent Gaussians, which eventually determines the expected number of Gaussians per unit volume. In other words, the expected distance between two adjacent Gaussians determines the expected size of local region per unit set of Gaussians.
>
> Meanwhile, we can approximate the distance between two adjacent Gaussians as the sum of their scales assuming there is no empty space in between. In 3DGS, what determines the average size of a Gaussian is $\tau\_{s}$, the hyperparameter in Adaptive Density Control algorithm in 3DGS, where a Gaussian is split if its maximum scale is greater than $\tau\_{s}$. Thus, the expected distance between two Gaussians cannot be greater than $2 \cdot \tau\_{s}$ due to the splitting algorithm.
>
> Based on the common intuition that optimal $K$ is inversely proportional to the size of local region, we may conclude that $K$ is also be inversely proportional to $\tau\_{s}$. From experiments, we learned that $\tau\_{s}$ smaller than the value proposed by the original 3DGS brought small performance improvements with optimal $K$ as `7` yet created too much Gaussians and often yielded OOM errors. Meanwhile, larger $\tau\_{s}$ decreased the performance regardless of $K$, since the scene became too sparse. Thus, we sticked to the default value for $\tau\_{s}$, and found the correspondingly optimal $K$ as `5`.
>
> >
> > ## $K$ for varying surface complexity
> >
>
> We strongly agree with the reviewer that there may exist an optimal $K$ for varying complexity. To make an assessment as best as possible, we counted the number of scene that performs the best for different $K$, assuming that scenes in DTU have varying complexity. For results, out of $15$ scenes in DTU, $1, 8, 4, 1$ and $1$ of them achieved the best results when $K$ is $4, 5, 6, 7$ and $8$, respectively. Thus, `5` is the best starting point for $K$ with the search range form `4` to `8`.
>
> >
> > ## Can the introduced regularization improve vanilla 3DGS?
> >
>
> Thank you for asking this question. To get to the point straight, we empirically learned that our regularization on vanilla 3DGS had a noticable impact if it is solely used without parameterization. Specifially, we experimented on 3DGS vs. 3DGS + our regularization ($\mathcal{L}\_{\text{eigval}}$ + $\mathcal{L}\_{\text{eigvec}}$) only. Note that for both methods, we used the improved densification criteria as in Eq.(2) of our paper. From experiments, we learned that without and with our regualrization on 3DGS yields Chamfer Distance of $1.305$ and $1.281$, respectively.
>
> >
> >## CD improvement of 0.02 is too small?
> >
>
> Please kindly refer to the answer `1. On seemingly negligible improvement in quantitative geometry measures.` from the general response!
>
> >
> >## Training is too slow?
> >
>
> Please kindly refer to the answer `3. Analysis and discussion on training time` from the general response!

---

### Official Review · Reviewer_YMCd · 2024-11-03

**Soundness:** 3
**Presentation:** 3
**Contribution:** 3
**Rating:** 6
**Confidence:** 4

**Summary:**

The paper proposes to improve the geometry quality of GS reconstruction by incorperating an inductive bias in GS covariance parameterization. The author claim that the proposed method reaches the state-of-art.

**Strengths:**

1) The presentation and writing is clear and easy to understand.
2) The proposed method regard every splat is not independent in cov computing process, which is intuitively reasonable.
3) The sparsity-aware density control strategy is interesting and useful.
4) The geometries of the demonstrated samples are impressive.

**Weaknesses:**

1) As shown in Table 1, although the proposed method maintains the faster training speed compared with the implicit reconstruction method, it takes longer time to train compared all listed explicit methods. However, the fast training  is one of the main advantages of GS. The proposed method need over 6 times of training time compared with the original GS.
2) The proposed method uses adjacent Gaussians for covariance parameterization. In different scenes with different scales, how to select K neighbours as "adjacent Gaussians"  is a question.
3) No failure cases are discussed.

**Questions:**

1) Have you investigated that which part of the proposed GS variant cost too much time in training?
2) Any ablations on the K neighbours of Gaussian for cov parameterization?
3) Any failure cases?
4) Since the GS pointclouds are sparse, with shape of "Splat". The surface reconstruction quality may lies in any part of the whole chain of reconstruction. For example, how do you integrate depth to TSDF volume? Do the listed methods for comparison use same method for mesh extraction?

---

> ### Author Response · Authors · 2024-11-19
>
> **We thank the reviewer for acknowledging our work. We also think that the reviewer's perspective on having to discuss the surface reconstruction quality along with TSDF fusion is an important question to ask. We will make sure to reflect the following discussions in appendix.**
>
>
> >
> > ## Discussing the surface reconstruction quality through the whole chain of reconstruction.
> >
>
> As the reviewer pointed out, it is important to understand the whole procedure, including the eventual surface mesh reconstruction given the learned scene representations, in order to throughly evaluate the surface reconstruction quality of different inverse rendering-based methods.
>
> To construct a TSDF volume, we first define the 3D world as a grid of voxels, where each voxel stores the signed distance to the nearest surface. Depth rasterized on a pixel is calculated using the ray intersection with the Gaussian whose accumulated opacity is `0.5`. 3D points from the depth image are mapped to the world coordinate system using the camera pose. These points update the TSDF volume, and new measurements are fused using a weighted average, where depths closer to the camera center are given higher weight. Marching Cubes is then applied to extract meshes from the voxel grid. This method has widely been adapted by the recent surface reconsturuction methods, which is why we followed this convention without modification. We apply the same procedure for all methods for surface extraction.
>
> As can be understood by the TSDF volume fusion algorithm, one of the keys for accurate surface reconstruction is the accurate depth map.
>
> Yet for 3DGS-based methods, depth becomes inaccurate if the covariance is not aligned and slanted with respect to the underlying surface. In other words, there exists a gap between the ray-Gaussian intersection and the underlying surface proportional to the slanted angle.
> This is why our SVD-based parameterization yields more accurate depth maps, because it aligns the Gaussian covariance to the underlying surface, making the slanted angle intuitively 0.
>
> The reviewer may also refer to our answer to the reviewer `XitP` on the comparison with VCR-GauS, which also discusses how the covariance misaligned to the surface creates imprecise depth rendering.
>
> >
> > ## Which part of our method cost too much time in training?
> >
>
> Please kindly refer to the answer `3. Analysis and discussion on training time` from the general response!
>
> >
> > ## Ablation on K neighbors for cov parameterization
> >
>
> Please kindly refer to the answer `4. Ablation on K` from the general response, and optionally our answers to the reviewer `amX1` on `K for varying size of local regions` and `K for varying surface complexity`!
>
> >
> > ## Failure cases
> >
> Please kindly refer to the answer `2. Failure case & limitations` from the general response!

---

### Official Review · Reviewer_XitP · 2024-11-06

**Soundness:** 3
**Presentation:** 3
**Contribution:** 3
**Rating:** 8
**Confidence:** 5

**Summary:**

This paper introduces a method called IBGS that "injects Inductive Bias to 3D Gaussian Splatting". The key idea is computing normals from the distribution of neighboring densities instead of from independently trainable Gaussian covariances. This paper also proposes geometry regularization methods to help form smooth local surface. Experiences on multiple datasets show that the proposed method achieves state of the art in surface reconstruction tasks among 3DGS-based methods while maintaining faster training time compared to implicit neural representation-based methods.

**Strengths:**

1. The key idea is well-motivated and novel. 3DGS-based methods have been a promising direction for faster surface reconstruction algorithms. A few papers have been trying to improve the normal calculation from 3D Gaussians. This paper brings a new perspective and proposes a novel normal calculation approach, aiming to predict more coherent surface. The new calculation also comes with novel regularization techniques for the normals.

2. Quantitative and qualitative results on multiple datasets demonstrate the effectiveness of the proposed method -- it achieves new state of the art in surface reconstruction among 3DGS-based methods while maintaining faster training time compared to implicit neural representation-based methods.

3. The paper is overall well-written and easy to follow. Implementation details sufficiently discussed.

**Weaknesses:**

1. The rationale behind calculating normals form neighboring Gaussians is well explained. Some of the regularization / splitting techniques used in this paper look very similar to the techniques applied in methods that optimize individual Gaussians. Could the authors help explain the difference and share some ablation studies (e.g., if they happen to have done these comparisons) for readers to better understand the effects of these techniques?

(1) This paper minimizes the smallest Eigen value lambda_3 (Sec. 3.3. 1, Eq. 8), this looks similar to (Sec. 3.2.1, Eq. 7) of NeuSG [A] that minimizes the smallest component of each Gaussian's three scaling factors.

(2) The sparsity-aware adaptive density control of this paper (Sec. 3.4) seems very similar to VCR-GauS [B] Sec. 3.4 that tries to split large Gaussians in the textureless areas by placing new Gaussians that evenly divide the maximum scale of the old Gaussian.

[A] Chen et al. Neusg: Neural implicit surface reconstruction with 3d gaussian splatting guidance. (already in the references)
[B] Chen et al. VCR-GauS: View Consistent Depth-Normal Regularizer for Gaussian Surface Reconstruction. NeurIPS 2024.

**Questions:**

The improved surface quality is well illustrated in the results. I was wondering if there are some remaining failure cases / limitations of the proposed approach. For example, the training speed seems a bit slower than other 3DGS-based methods, is it caused by the overhead of computing properties among K nearest neighbors?

---

> ### Author Response · Authors · 2024-11-19
>
> **We thank the reviewer for acknowledging our work, and most of all for pointing out important works for comparison.**
>
> >
> > ## Gaussian flattening loss vs. Smallest eigenvalue minimization loss
> >
>
> We thank the reviewer for contemplating the difference between scale minimization loss, $\mathcal{L}\_{\text{s}} = \text{min}(s_1, s_2, s_3)$, proposed by NeuSG **[C]** and our eigenvalue minimization loss, $\mathcal{L}\_{\text{eigval}} = \text{min}(\lambda_1, \lambda_2, \lambda_3) = \lambda_3$.
>
> Getting to the point straight, $\mathcal{L}\_{\text{s}}$ is designed to flatten the shape of a single Gaussian, whereas $\mathcal{L}\_{\text{eigval}}$ induces adjacent Gaussians to be located near a local surface.
> In other words, shape of a Gaussian is not necessarily flat under $\mathcal{L}\_{\text{eigval}}$, and neighboring Gaussians are not necessarily distributed on a local plane under $\mathcal{L}\_{\text{s}}$.
>
> The cause of the diffence are the distinct propoerties of $s_i$ and $\lambda_i$. Specifically, $s_i$ represents the scale of a ***single Gaussian***, whereas $\lambda_i$ represents the variance of distribution of ***multiple neighboring Gaussians***, as formulated in Eq.(4) and Eq.(5) in our paper.
>
> Such difference naturally brings another question whether applying  $\mathcal{L}\_{\text{s}}$ and $\mathcal{L}\_{\text{eigval}}$ brings the benefits of the both, i.e., flat gaussians distributed on a local plane. This was in fact one of the options to improve our work during the research. However, we learned that only applying $\mathcal{L}\_{\text{eigval}}$ performs the best, as reported on the table below. We conjecture that applying both losses impose too much constraint on expressibility. Yet, it would be an interesting research topic to investigate another possible cause of this observation.
>
> ||$\mathcal{L}\_{\text{s}}$ | $\mathcal{L}\_{\text{eigval}}$ | $\mathcal{L}\_{\text{s}} + \mathcal{L}\_{\text{eigval}} $ |
> |:-:| :-: | :-: | :-: |
> |CD |0.689 | 0.674 | 0.685 |
>
> >
> > ## Comparison with VCR-GauS
> >
>
> * **Comapring the densification algorithms**
>
> We thank the reviewer for bringing up the densification algorithm from VCR-GauS **[D]** for comparison with our Sparsity-aware ADC (SADC). To make a thorough comparison, we read the code release of VCR-GauS to claify some ambiguous points from the paper on how VCR-GauS finds the "surface" that defines a plane that the splitted Gaussians will be located into (the reviewer may refer to Figure 4-(a) of VCR-GauS paper). From the code, we learned that the Gaussians are simply splitted along the axis of maximum scale. We conjecture VCR-GauS to assume that the Gaussians are splitted along the surface because they split Gaussians that intersect with the camera rays for the first time, which are the most likely to be located on surface. However, $R$ is still fully and independently trainable for Gaussians in VCR-GauS, meaning that the splitting direction is still not aware of the locations of adjacent Gaussians.
>
> Meanwhile, our SADC explicitly samples new Gaussians between the neighbooring Gaussians. Thus, we believe that SADC makes stricter implementation on sampling new Gaussians on a surface defined by the distribution of neighboring Gaussians.
>
> * **Discussion on how covariance aligned to surface reduces depth error**
>
> In addition, the intuition behind introducing the densification algorithm in VCR-GauS is to minimize the gap between the depth calculated from ray-Gaussian intersection and the underlying surface, which is approximated as $s \cdot \text{sin}(\theta)$, where $s$ is the Gaussian scale and $\theta$ is the angle between the axis of the scale and the underlying surface (again, please refer to Figure 4-(a) of VCR-GauS paper). Our work, on the other hand, is designed to systematically remove such gap by explicitly parameterizing the Gaussian orientation ***using*** the surface, making $\theta = 0$ intuitively. Thus, comparing the objectives behind the two densification algorithm highlights the fundamental difference between the two models in wider perspective.
>
> >
> > ## ON failure cases / limitations
> >
>
> Please kindly refer to the answer `2. Failure case & limitations` from the general response!
>
> >
> > ## On training speed
> >
>
> Please kindly refer to the answer `3. Analysis and discussion on training time` from the general response!
>
> >
> > ***Reference***
> >
>
> **[C]** Chen et al. NeuSG: Neural implicit surface reconstruction with 3d gaussian splatting guidance.
>
> **[D]** Chen et al. VCR-GauS: View Consistent Depth-Normal Regularizer for Gaussian Surface Reconstruction. NeurIPS 2024.

---

### Author Response · Authors · 2024-11-19
**General Response**

We thank the reviewers for their insightful and valuable feedback.

**Especially, we appreciate all four reviewers for recognizing:**
  * Novelty and sanity of our approach
  * Clear demonstration in qualitative improvements on multiple datasets
  * Clear presentation

Below, we summarize some common concerns from the reviewers.

>
> ## **(1) On seemingly negligible improvement in quantitative measures.**
>

We totally understand that 0.02 gain in Chamfer Distance (CD) may seem negligible if viewed by numbers only. However, the improvement becomes more obvious when the equivalent scene is observed qualitatively. For instance, CD of the scene (scan ID `65` of DTU dataset) in Figure 1 of our paper trained with RaDe-GS and our method is `0.76` and `0.73`, respectively, yielding an improvement of `0.03`. Although this number may also seem negligible, the following visual difference is significant.

Also, please zoom in for detailed comparisons on parts that are not highlighted as well, from which smoother surfaces can be observed from our method over RaDe-GS. Such sophisticated improvements make reasonable impacts on users, but can easily be underestimated when perceived in quantitative measures.

>
> ## **(2) Failure case & limitations**
>

One limitation of our method is that it does not work on scenes where objects of interest are located far away from the cameras or unbounded scenes. It is due to the nature of perspective projection-based inverse rendering on descrete representations, where Gaussians gets sparser when they get farther away from camera. For instance, consider two rays from two adjacent pixels, respectively. Also assume that there are two Gaussians, each of which intersect one of the rays and cover the following pixel. Then, the farther away these two Gaussians are from the camera, the larger the distance would be between the two Gaussians. Since kNN will cover larger areas on sparser Gaussians, it is hard to assume the locality of the plane formed by these Gaussians.

A possible solution to this limitation is to find neighbors using radius-based ball query **[A]**, so that neighbors that are too far away from the query can be rejcted. However, we did not experiment on ball query as there exists no implementation that is tensor-friendly (because ball query returns variable number of neighbors) and efficient compared to GPU-based library for kNN such as Faiss **[B]**. We will make sure to add these limitations in our paper.


>
> ## **(3) Analysis and discussion on training time**
>

* ***Training time analysis***

Compared to RaDe-GS, extra forward and backward process introduced by SVD and kNN were two key factors of increased training time. We made experimental analysis on how each of them contribute to the training time. We experimented on DTU Dataset with a single RTX 3090 GPU, and computed average training time for all scenes.

First, applying SVD only increase training time from `16.2` min to `43.8` min, from which we can do a simple math to conclude that SVD contributed by `27.6` min. Since our full implementation took `91.2` min, kNN can be estimated to contribute by $91.2 - 43.8 = $ `47.4` min. By percentage, SVD and kNN contributes `30.2%` and `52.0%` of the training time, respectively. Note that kNN is conducted for every $100$ iteration throughout the whole training steps of $30\text{k}$ iterations.

* ***Plans on efficient implementation for faster training***

One possible way to reduce training time to conduct SVD on points in view-frustum only for every iteration. For kNN, we can keep track of the change for every kNN graph update, and stop updating when few of the neighbors change. We believe that these simple engineering tricks can noticably reduce the training time, and planning to make the implementation upon the release of the code.


However, we would like to point out that even without these tricks, training time is already similar to the recent 3DGS-based methods such as SuGaR and GoF. We also hope that future research can investigate on other fundamentally novel approaches to improve training time while keeping the inductive bias.

>
> ## (4) **Ablation on $K$**
>

We share the ablation results on $K$ on DTU Dataset. From the results, we learned that $K=5$ yields the best results in Chamfer Distance (CD). We have edited the experimental result in the appendix section of our manuscript.

  |K|2|3|4|5|6|7|8|9|10|
  |-|-|-|-|-|-|-|-|-|-|
  |CD| 0.701 | 0.709 | 0.684 | ***0.674*** | 0.678 | 0.686 | 0.684  | 0.688 | 0.692 |

>
> ***Reference***
>

  **[A]** Qi, Charles Ruizhongtai, et al. "Pointnet++: Deep hierarchical feature learning on point sets in a metric space." NeurIPS, 2017.

  **[B]** Johnson, et al. "Billion-scale similarity search with GPUs." IEEE Transactions on Big Data, 2019.

---

### Meta-Review · Area_Chair_CgdB · 2024-12-20

**Metareview:**

This paper describes a method for improving the geometry quality of Gaussian splatting by incorporating the inductive bias in reconstruction. Unlike previous methods that learn a surface normal for each Gaussian, the paper proposes to learn normals using the distribution of neighboring Gaussians. The strength of the work is the introduction of the new idea in the surface normal (or Gaussian covariance) computation scheme, and the quality of the result. On the other hand, the chief weakness was that the proposed surface normal computation is not adequately grounded. There was an extensive discussion about this respect during the reviewer-author discussion phase. Unfortunately, the concern was not clarified during the discussion. As a result, there were mixed opinions about the paper. The AC agreed with the opinion that the justification for the new surface normal computation needed to be fully explained because it was the key motivation for the work. Due to the lack of clear justification, we reached this recommendation.

**Additional Comments On Reviewer Discussion:**

There was an extensive discussion about the motivation/correctness about the new surface normal computation method in 3DGS. It was the central topic for this paper. Unfortunately, the reviewer's and authors' opinions did not converge. The AC agreed with the reviewer's point that the proposed surface normal computation was not adequately grounded, which led to the lack of motivation for the development of the method. Although the reported quality score is good, it remained unclear what was the key enabler for this result.

---

### Decision · Program_Chairs · 2025-01-22

Reject